# A Sublinear-Time Spectral Clustering Oracle with Improved Preprocessing Time

**Ranran Shen**[*]
ranranshen@mail.ustc.edu.cn

**Pan Peng**[*]
ppeng@ustc.edu.cn

## Abstract

We address the problem of designing a sublinear-time spectral clustering oracle for graphs that exhibit strong clusterability. Such graphs contain $k$ latent clusters, each characterized by a large inner conductance (at least $\varphi$) and a small outer conductance (at most $\varepsilon$). Our aim is to preprocess the graph to enable clustering membership queries, with the key requirement that both preprocessing and query answering should be performed in sublinear time, and the resulting partition should be consistent with a $k$-partition that is close to the ground-truth clustering. Previous oracles have relied on either a $\text{poly}(k) \log n$ gap between inner and outer conductances or exponential (in $k/\varepsilon$) preprocessing time. Our algorithm relaxes these assumptions, albeit at the cost of a slightly higher misclassification ratio. We also show that our clustering oracle is robust against a few random edge deletions. To validate our theoretical bounds, we conducted experiments on synthetic networks.

## 1 Introduction

Graph clustering is a fundamental task in the field of graph analysis. Given a graph $G = (V, E)$ and an integer $k$, the objective of graph clustering is to partition the vertex set $V$ into $k$ disjoint clusters $C_1, \ldots, C_k$. Each cluster should exhibit tight connections within the cluster while maintaining loose connections with the other clusters. This task finds applications in various domains, including community detection [32, 12], image segmentation [11] and bio-informatics [29].

However, global graph clustering algorithms, such as spectral clustering [27], modularity maximization [26], density-based clustering [10], can be computationally expensive, especially for large datasets. For instance, spectral clustering is a significant algorithm for solving the graph clustering problem, which involves two steps. The first step is to map all the vertices to a $k$-dimensional Euclidean space using the Laplacian matrix of the graph. The second step is to cluster all the points in this $k$-dimensional Euclidean space, often employing the $k$-means algorithm. The time complexity of spectral clustering, as well as other global clustering algorithms, is $\text{poly}(n)$, where $n = |V|$ denotes the size of the graph. As the graph size increases, the computational demands of these global clustering algorithms become impractical.

Addressing this challenge, an effective approach lies in the utilization of local algorithms that operate within sublinear time. In this paper, our primary focus is on a particular category of such algorithms designed for graph clustering, known as *sublinear-time spectral clustering oracles* [30, 14]. These algorithms consist of two phases: the preprocessing phase and the query phase, both of which can be executed in sublinear time. During the preprocessing phase, these algorithms sample a subset of vertices from $V$, enabling them to locally explore a small portion of the graph and gain insights into its cluster structure. In the query phase, these algorithms utilize the cluster structure learned during the preprocessing phase to respond to WHICHCLUSTER$(G, x)$ queries. The resulting partition

---

[*]School of Computer Science and Technology, University of Science and Technology of China, Hefei, China. Corresponding author: Pan Peng.

37th Conference on Neural Information Processing Systems (NeurIPS 2023).

defined by the output of WHICHCLUSTER$(G, x)$ should be consistent with a $k$-partition that is close to the ground-truth clustering.

We study such oracles for graphs that exhibit strong clusterability, which are graphs that contain $k$ latent clusters, each characterized by a large inner conductance (at least $\varphi$) and a small outer conductance (at most $\varepsilon$). Let us assume $\varphi > 0$ is some constant. In [30] (see also [8]), a robust clustering oracle was designed with preprocessing time approximately $O(\sqrt{n} \cdot \text{poly}(\frac{k \log n}{\varepsilon}))$, query time approximately $O(\sqrt{n} \cdot \text{poly}(\frac{k \log n}{\varepsilon}))$, misclassification error (i.e., the number of vertices that are misclassified with respect to a ground-truth clustering) approximately $O(kn\sqrt{\varepsilon})$. The oracle relied on a $\text{poly}(k) \log n$ gap between inner and outer conductance. In [14], a clustering oracle was designed with preprocessing time approximately $2^{\text{poly}(\frac{k}{\varepsilon})}\text{poly}(\log n) \cdot n^{1/2+O(\varepsilon)}$, query time approximately $\text{poly}(\frac{k \log n}{\varepsilon}) \cdot n^{1/2+O(\varepsilon)}$, misclassification error $O(\log k \cdot \varepsilon)|C_i|$ for each cluster $C_i, i \in [k]$ and it takes approximately $O(\text{poly}(\frac{k}{\varepsilon}) \cdot n^{1/2+O(\varepsilon)} \cdot \text{poly}(\log n))$ space. This oracle relied on a $\log k$ gap between inner and outer conductance.

One of our key contributions in this research is a new sublinear-time spectral clustering oracle that offers enhanced preprocessing efficiency. Specifically, we introduce an oracle that significantly reduces both the preprocessing and query time, performing in $\text{poly}(k \log n) \cdot n^{1/2+O(\varepsilon)}$ time and reduces the space complexity, taking $O(\text{poly}(k) \cdot n^{1/2+O(\varepsilon)} \cdot \text{poly}(\log n))$ space. This approach relies on a $\text{poly}(k)$ gap between the inner and outer conductances, while maintaining a misclassification error of $O(\text{poly}(k) \cdot \varepsilon^{1/3})|C_i|$ for each cluster $C_i, i \in [k]$. Moreover, our oracle offers practical implementation feasibility, making it well-suited for real-world applications. In contrast, the clustering oracle proposed in [14] presents challenges in terms of implementation (mainly due to the exponential dependency on $k/\varepsilon$).

We also investigate the sensitivity of our clustering oracle to edge perturbations. This analysis holds significance in various practical scenarios where the input graph may be unreliable due to factors such as privacy concerns, adversarial attacks, or random noises [31]. We demonstrate the robustness of our clustering oracle by showing that it can accurately identify the underlying clusters in the resulting graph even after the random deletion of one or a few edges from a well-clusterable graph.

**Basic definitions.** Graph clustering problems often rely on conductance as a metric to assess the quality of a cluster. Several recent studies ([8, 9, 30, 14, 22]) have employed conductance in their investigations. Hence, in this paper, we adopt the same definition to characterize the cluster quality. We state our results for $d$-regular graphs for some constant $d \geq 3$, though they can be easily generalized to graphs with maximum degree at most $d$ (see Appendix B).

**Definition 1.1** (Inner and outer conductance)**.** Let $G = (V, E)$ be a $d$-regular $n$-vertex graph. For a set $S \subseteq C \subseteq V$, we let $E(S, C\backslash S)$ denote the set of edges with one endpoint in $S$ and the other endpoint in $C\backslash S$. The *outer conductance of a set* $C$ is defined to be $\phi_{\text{out}}(C, V) = \frac{|E(C,V\backslash C)|}{d|C|}$. The *inner conductance of a set* $C$ is defined to be $\phi_{\text{in}}(C) = \min\limits_{S \subseteq C, 0 < |S| \leq \frac{|C|}{2}} \phi_{\text{out}}(S, C) = \min\limits_{S \subseteq C, 0 < |S| \leq \frac{|C|}{2}} \frac{|E(S,C\backslash S)|}{d|S|}$ if $|C| > 1$ and one otherwise. Specially, *the conductance of graph $G$ is* defined to be $\phi(G) = \min\limits_{C \subseteq V, 0 < |C| \leq \frac{n}{2}} \phi_{\text{out}}(C, V)$.

Note that based on the above definition, for a cluster $C$, the smaller the $\phi_{\text{out}}(C, V)$ is, the more loosely connected with the other clusters and the bigger the $\phi_{\text{in}}(C)$ is, the more tightly connected within $C$. For a high quality cluster $C$, we have $\phi_{\text{out}}(C, V) \ll \phi_{\text{in}}(C) \leq 1$.

**Definition 1.2** ($k$-partition)**.** Let $G = (V, E)$ be a graph. *A $k$-partition of $V$ is a collection of disjoint subsets $C_1, \ldots, C_k$ such that $\cup_{i=1}^{k} C_i = V$.*

Based on the above, we have the following definition of clusterable graphs.

**Definition 1.3** ($(k, \varphi, \varepsilon)$-clustering)**.** Let $G = (V, E)$ be a $d$-regular graph. A $(k, \varphi, \varepsilon)$-*clustering of* $G$ is a $k$-partition of $V$, denoted by $C_1, \ldots, C_k$, such that for all $i \in [k]$, $\phi_{\text{in}}(C_i) \geq \varphi$, $\phi_{\text{out}}(C_i, V) \leq \varepsilon$ and for all $i, j \in [k]$ one has $\frac{|C_i|}{|C_j|} \in O(1)$. $G$ is called a $(k, \varphi, \varepsilon)$-*clusterable graph* if there exists a $(k, \varphi, \varepsilon)$-clustering of $G$.

**Main results.** Our main contribution is a sublinear-time spectral clustering oracle with improved preprocessing time for $d$-regular $(k, \varphi, \varepsilon)$-clusterable graphs. We assume query access to the adjacency list of a graph $G$, that is, one can query the $i$-th neighbor of any vertex in constant time.

**Theorem 1.** *Let $k \geq 2$ be an integer, $\varphi \in (0, 1)$. Let $G = (V, E)$ be a $d$-regular $n$-vertex graph that admits a $(k, \varphi, \varepsilon)$-clustering $C_1, \ldots, C_k$, $\frac{\varepsilon}{\varphi^2} \ll \frac{\gamma^3}{k^{\frac{9}{2}} \cdot \log^3 k}$ and for all $i \in [k]$, $\gamma \frac{n}{k} \leq |C_i| \leq \frac{n}{\gamma k}$, where $\gamma$ is a constant that is in $(0.001, 1]$. There exists an algorithm that has query access to the adjacency list of $G$ and constructs a clustering oracle in $O(n^{1/2+O(\varepsilon/\varphi^2)} \cdot \text{poly}(\frac{k \log n}{\gamma \varphi}))$ preprocessing time and takes $O(n^{1/2+O(\varepsilon/\varphi^2)} \cdot \text{poly}(\frac{k \log n}{\gamma}))$ space. Furthermore, with probability at least $0.95$, the following hold:*

1. *Using the oracle, the algorithm can answer any WHICHCLUSTER query in $O(n^{1/2+O(\varepsilon/\varphi^2)} \cdot \text{poly}(\frac{k \log n}{\gamma \varphi}))$ time and a WHICHCLUSTER query takes $O(n^{1/2+O(\varepsilon/\varphi^2)} \cdot \text{poly}(\frac{k \log n}{\gamma}))$ space.*

2. *Let $U_i := \{x \in V : \text{WHICHCLUSTER}(G, x) = i\}, i \in [k]$ be the clusters recovered by the algorithm. There exists a permutation $\pi : [k] \to [k]$ such that for all $i \in [k]$, $|U_{\pi(i)} \triangle C_i| \leq O(\frac{k^{\frac{3}{2}}}{\gamma} \cdot (\frac{\varepsilon}{\varphi^2})^{1/3})|C_i|$.*

Specifically, for every graph $G = (V, E)$ that admits a $k$-partition $C_1, \ldots, C_k$ with *constant* inner conductance $\varphi$ and outer conductance $\varepsilon \ll O(\frac{1}{\text{poly}(k)})$, our oracle has preprocessing time $\approx n^{1/2+O(\varepsilon)} \cdot \text{poly}(k \log n)$, query time $\approx n^{1/2+O(\varepsilon)} \cdot \text{poly}(k \log n)$, space $\approx O(n^{1/2+O(\varepsilon/\varphi^2)} \cdot \text{poly}(k \log n))$ and misclassification error $\approx O(\text{poly}(k) \cdot \varepsilon^{1/3})|C_i|$ for each cluster $C_i, i \in [k]$. In comparison to [30], our oracle relies on a smaller gap between inner and outer conductance (specifically $O(\text{poly}(k) \log n)$). In comparison to [14], our oracle has a smaller preprocessing time and a smaller space at the expense of a slightly higher misclassification error of $O(\text{poly}(k) \cdot \varepsilon^{1/3})|C_i|$ instead of $O(\log k \cdot \varepsilon)|C_i|$ and a slightly worse conductance gap of $\varepsilon \ll O(\varphi^2/\text{poly}(k))$ instead of $\varepsilon \ll O(\varphi^3/\log(k))$. It's worth highlighting that our space complexity significantly outperforms that of [14] (i.e., $O(n^{1/2+O(\varepsilon/\varphi^2)} \cdot \text{poly}(\frac{k}{\varepsilon} \cdot \log n))$), particularly in cases where $k$ is fixed and $\varepsilon$ takes on exceptionally small values, such as $\varepsilon = \frac{1}{n^c}$ for sufficiently small constant $c > 0$, since the second term in our space complexity does *not* depend on $\varepsilon$ in comparison to the one in [14].

Another contribution of our work is the verification of the robustness of our oracle against the deletion of one or a few random edges. The main idea underlying the proof is that a well-clusterable graph is still well-clusterable (with a slightly worse clustering quality) after removing a few random edges, which in turn is built upon the intuition that after removing a few random edges, an expander graph remains an expander. For the complete statement and proof of this claim, we refer to Appendix E.

**Theorem 2** (Informal; Robust against random edge deletions). *Let $c > 0$ be a constant. Let $G_0$ be a graph satisfying the similar conditions as stated in Theorem 1. Let $G$ be a graph obtained from $G_0$ by randomly deleting $c$ edges. Then there exists a clustering oracle for $G$ with the same guarantees as presented in Theorem 1.*

## 1.1 Related work

Sublinear-time algorithms for graph clustering have been extensively researched. Czumaj et al. [8] proposed a property testing algorithm capable of determining whether a graph is $k$-clusterable or significantly far from being $k$-clusterable in sublinear time. This algorithm, which can be adapted to a sublinear-time clustering oracle, was later extended by Peng [30] to handle graphs with noisy partial information through a robust clustering oracle. Subsequent improvements to both the testing algorithm and the oracle were introduced by Chiplunkar et al. [6] and Gluchowski et al. [14]. Recently, Kapralov et al. [16, 17] presented a hierarchical clustering oracle specifically designed for graphs exhibiting a pronounced hierarchical structure. This oracle offers query access to a high-quality hierarchical clustering at a cost of $\text{poly}(k) \cdot n^{1/2+O(\gamma)}$ per query. However, it is important to note that their algorithm does not provide an oracle for flat $k$-clustering, as considered in our work, with the same query complexity. Sublinear-time clustering oracles for signed graphs have also been studied recently [25].

The field of *local graph clustering* [33, 1, 3, 2, 34, 28] is also closely related to our research. In this framework, the objective is to identify a cluster starting from a given vertex within a running time

that is bounded by the size of the output set, with a weak dependence on $n$. Zhu et al. [34] proposed a local clustering algorithm that produces a set with low conductance when both inner and outer conductance are used as measures of cluster quality. It is worth noting that the running times of these algorithms are sublinear only if the target set's size (or volume) is small, for example, at most $o(n)$. In contrast, in our setting, the clusters of interest have a minimum size that is $\Omega(n/k)$.

Extensive research has been conducted on fully or partially recovering clusters in the presence of noise within the "global algorithm regimes". Examples include recovering the planted partition in the *stochastic block model* with modeling errors or noise [4, 15, 24, 21], *correlation clustering* on different ground-truth graphs in the *semi-random* model [23, 5, 13, 20], and graph partitioning in the *average-case* model [18, 19, 20]. It is important to note that all these algorithms require at least linear time to run.

## 2   Preliminaries

Let $G = (V, E)$ denote a $d$-regular undirected and unweighted graph, where $V := \{1, \ldots, n\}$. Throughout the paper, we use $i \in [n]$ to denote $1 \leq i \leq n$. For any two vectors $\mathbf{x}, \mathbf{y} \in \mathbb{R}^n$, we let $\langle \mathbf{x}, \mathbf{y} \rangle = \mathbf{x}^T \mathbf{y}$ denote the dot product of $\mathbf{x}$ and $\mathbf{y}$. For a graph $G$, we let $A \in \mathbb{R}^{n \times n}$ denote the adjacency matrix of $G$ and let $D \in \mathbb{R}^{n \times n}$ denote a diagonal matrix with $D(i, i) = \deg(i)$, where $\deg(i)$ is the degree of vertex $i, i \in [n]$. We denote with $L$ the normalized Laplacian of $G$ where $L = D^{-1/2}(D - A)D^{-1/2} = I - \frac{A}{d}$. For $L$, we use $0 \leq \lambda_1 \leq \ldots \leq \lambda_n \leq 2$ [7] to denote its eigenvalues and we use $u_1, \ldots, u_n \in \mathbb{R}^n$ to denote the corresponding eigenvectors. Note that the corresponding eigenvectors are not unique, in this paper, we let $u_1, \ldots, u_n$ be an orthonormal basis of eigenvectors of $L$. For any two sets $S_1$ and $S_2$, we let $S_1 \triangle S_2$ denote the symmetric difference between $S_1$ and $S_2$.

Due to space constraints, we present only the key preliminaries here; the complete preliminaries will be presented in Appendix A.

Our algorithms in this paper are based on the properties of the dot product of spectral embeddings, so we also need the following definition.

**Definition 2.1** (Spectral embedding). For a graph $G = (V, E)$ with $n = |V|$ and an integer $2 \leq k \leq n$, we use $L$ denote the normalized Laplacian of $G$. Let $U_{[k]} \in \mathbb{R}^{n \times k}$ denote the matrix of the bottom $k$ eigenvectors of $L$. Then for every $x \in V$, *the spectral embedding of $x$,* denoted by $f_x \in \mathbb{R}^k$, is the $x$-row of $U_{[k]}$, which means $f_x(i) = u_i(x), i \in [k]$.

**Definition 2.2** (Cluster centers). Let $G = (V, E)$ be a $d$-regular graph that admits a $(k, \varphi, \varepsilon)$-clustering $C_1, \ldots, C_k$. The *cluster center* $\mu_i$ of $C_i$ is defined to be $\mu_i = \frac{1}{|C_i|} \sum_{x \in C_i} f_x, i \in [k]$.

The following informal lemma shows that the dot product of two spectral embeddings can be approximated in $\widetilde{O}(n^{1/2 + O(\varepsilon/\varphi^2)} \cdot \text{poly}(k))$ time.

**Lemma 2.1** (Informal; Theorem 2, [14]). *Let $G = (V, E)$ be a $d$-regular graph that admits a $(k, \varphi, \varepsilon)$-clustering $C_1, \ldots, C_k, \varepsilon \leq \frac{\varphi^2}{10^5}$. Then for every pair of vertices $x, y \in V$, SPECTRALDOTPRODUCT$(G, x, y, 1/2, \xi, \mathcal{D})$ computes an output value $\langle f_x, f_y \rangle_{\text{apx}}$ in $\widetilde{O}(n^{1/2 + O(\varepsilon/\varphi^2)} \cdot \text{poly}(k))$ time such that with high probability:* $\left| \langle f_x, f_y \rangle_{\text{apx}} - \langle f_x, f_y \rangle \right| \leq \frac{\xi}{n}$.

For the completeness of this paper, we will show the formal Lemma 2.1 and algorithm SPECTRALDOTPRODUCT$(G, x, y, 1/2, \xi, \mathcal{D})$ in Appendix C.

## 3   Spectral clustering oracle

### 3.1   Our techniques

We begin by outlining the main concepts of the spectral clustering oracle presented in [14]. Firstly, the authors introduce a sublinear time oracle that provides dot product access to the spectral embedding of graph $G$ by estimating distributions of short random walks originating from vertices in $G$ (as described in Lemma 2.1). Subsequently, they demonstrate that (1) the set of points corresponding to the spectral embeddings of all vertices exhibits well-concentrated clustering around the cluster center $\mu_i$ (refer to Definition 2.2), and (2) all the cluster centers are approximately orthogonal to each other. The clustering oracle in [14] operates as follows: it initially guesses the $k$ cluster centers from a set of

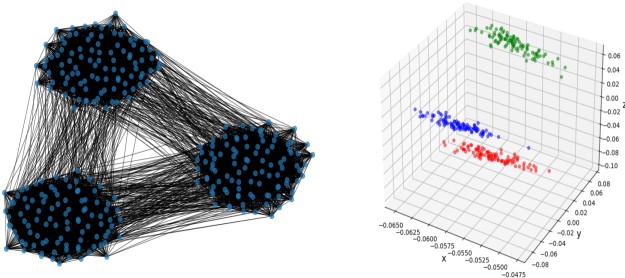

Figure 1: The angle between embeddings of vertices in the same cluster is small and the angle between embeddings of vertices in different clusters is close to orthogonal ($k = 3$).

poly$(k/\varepsilon)$ sampled vertices, which requires a time complexity of $2^{\text{poly}(k/\varepsilon)}n^{1/2+O(\varepsilon)}$. Subsequently, it iteratively employs the dot product oracle to estimate $\langle f_x, \mu_i \rangle$. If the value of $\langle f_x, \mu_i \rangle$ is significant, it allows them to infer that vertex $x$ likely belongs to cluster $C_i$.

Now we present our algorithm, which builds upon the dot product oracle in [14]. Our main insight is to avoid relying directly on cluster centers in our algorithm. By doing so, we can eliminate the need to guess cluster centers and consequently remove the exponential time required in the preprocessing phase described in [14]. The underlying intuition is as follows: if two vertices, $x$ and $y$, belong to the same cluster $C_i$, their corresponding spectral embeddings $f_x$ and $f_y$ will be close to the cluster center $\mu_i$. As a result, the angle between $f_x$ and $f_y$ will be small, and the dot product $\langle f_x, f_y \rangle$ will be large (roughly on the order of $O(\frac{k}{n})$). Conversely, if $x$ and $y$ belong to different clusters, their embeddings $f_x$ and $f_y$ will tend to be orthogonal, resulting in a small dot product $\langle f_x, f_y \rangle$ (close to 0). We prove that this desirable property holds for the majority of vertices in $d$-regular $(k, \varphi, \varepsilon)$-clusterable graphs (see Figure 1 for an illustrative example). Slightly more formally, we introduce the definitions of *good* and *bad* vertices (refer to Definition 3.1) such that the set of good vertices corresponds to the core part of clusters and each pair of good vertices satisfies the aforementioned property; the rest vertices are the bad vertices. Leveraging this property, we can directly utilize the dot product of spectral embeddings to construct a sublinear clustering oracle.

Based on the desirable property discussed earlier, which holds for $d$-regular $(k, \varphi, \varepsilon)$-clusterable graphs, we can devise a sublinear spectral clustering oracle. Let $G = (V, E)$ be a $d$-regular $(k, \varphi, \varepsilon)$-clusterable graph that possesses a $(k, \varphi, \varepsilon)$-clustering $C_1, \ldots, C_k$. In the preprocessing phase, we sample a set $S$ of $s$ vertices from $V$ and construct a similarity graph, denoted as $H$, on $S$. For each pair of vertices $x, y \in S$, we utilize the dot product oracle from [14] to estimate $\langle f_x, f_y \rangle$. If $x$ and $y$ belong to the same cluster $C_i$, yielding a large $\langle f_x, f_y \rangle$, we add an edge $(x, y)$ to $H$. Conversely, if $x$ and $y$ belong to different clusters, resulting in a $\langle f_x, f_y \rangle$ close to 0, we make no modifications to $H$. Consequently, only vertices within the same cluster $C_i(i \in [k])$ can be connected by edges. We can also establish that, by appropriately selecting $s$, the sampling set $S$ will, with high probability, contain at least one vertex from each $C_1, \ldots, C_k$. Thus, the similarity graph $H$ will have $k$ connected components, with each component corresponding to a cluster in the ground-truth. We utilize these $k$ connected components, denoted as $S_1, \ldots, S_k$, to represent $C_1, \ldots, C_k$.

During the query phase, we determine whether the queried vertex $x$ belongs to a connected component in $H$. Specifically, we estimate $\langle f_x, f_y \rangle$ for all $y \in S$. If there exists a unique index $i \in [k]$ for which $\langle f_x, f_u \rangle$ is significant (approximately $O(\frac{k}{n})$) for all $u \in S_i$, we conclude that $x$ belongs to cluster $C_i$, associated with $S_i$. If no such unique index is found, we assign $x$ a random index $i$, where $i \in [k]$.

### 3.2 The clustering oracle

Next, we present our algorithms for constructing a spectral clustering oracle and handling the WHICH-CLUSTER queries. In the preprocessing phase, the algorithm CONSTRUCTORACLE$(G, k, \varphi, \varepsilon, \gamma)$ learns the cluster structure representation of $G$. This involves constructing a similarity graph $H$ on a sampled vertex set $S$ and assigning membership labels $\ell$ to all vertices in $S$. During the query phase, the algorithm WHICHCLUSTER$(G, x)$ determines the clustering membership index to which vertex $x$ belongs. More specifically, WHICHCLUSTER$(G, x)$ utilizes the function SEARCHINDEX$(H, \ell, x)$

---

**Algorithm 1:** CONSTRUCTORACLE$(G, k, \varphi, \varepsilon, \gamma)$

---

1 Let $\xi = \frac{\sqrt{\gamma}}{1000}$ and let $s = \frac{10 \cdot k \log k}{\gamma}$

2 Let $\theta = 0.96(1 - \frac{4\sqrt{\varepsilon}}{\varphi})\frac{\gamma k}{n} - \frac{\sqrt{k}}{n}(\frac{\varepsilon}{\varphi^2})^{1/6} - \frac{\xi}{n}$

3 Sample a set S of $s$ vertices independently and uniformly at random from $V$

4 Generate a similarity graph $H = (S, \emptyset)$

5 Let $\mathcal{D} = $ INITIALIZEORACLE$(G, 1/2, \xi)$

6 **for** *any* $u, v \in S$ **do**

7      Let $\langle f_u, f_v \rangle_{\text{apx}} = $ SPECTRALDOTPRODUCTORACLE$(G, u, v, 1/2, \xi, \mathcal{D})$

8      **if** $\langle f_u, f_v \rangle_{\text{apx}} \geq \theta$ **then**

9          Add an edge $(u, v)$ to the similarity graph $H$

10 **if** *H has exactly k connected components* **then**

11      Label the connected components with $1, 2, \ldots, k$ (we write them as $S_1, \ldots, S_k$)

12      Label $x \in S$ with $i$ if $x \in S_i$

13      Return $H$ and the vertex labeling $\ell$

14 **else**

15      return **fail**

---

to check whether the queried vertex $x$ belongs to a unique connected component in $H$. If it does, SEARCHINDEX$(H, \ell, x)$ will return the index of the unique connected component in $H$.

The algorithm in preprocessing phase is given in Algorithm 1 CONSTRUCTORACLE$(G, k, \varphi, \varepsilon, \gamma)$.

See Appendix C for algorithm INITIALIZEORACLE and SPECTRALDOTPRODUCTORACLE invoked by CONSTRUCTORACLE$(G, k, \varphi, \varepsilon, \gamma)$.

Our algorithms used in the query phase are described in Algorithm 2 SEARCHINDEX$(H, \ell, x)$ and Algorithm 3 WHICHCLUSTER$(G, x)$.

---

**Algorithm 2:** SEARCHINDEX$(H, \ell, x)$

---

1 **for** *any vertex* $u \in S$ **do**

2      Let $\langle f_u, f_x \rangle_{\text{apx}} = $ SPECTRALDOTPRODUCTORACLE$(G, u, x, 1/2, \xi, \mathcal{D})$

3 **if** *there exists a unique index* $1 \leq i \leq k$ *such that* $\langle f_u, f_x \rangle_{\text{apx}} \geq \theta$ *for all* $u \in S_i$ **then**

4      return index $i$

5 **else**

6      return **outlier**

---

---

**Algorithm 3:** WHICHCLUSTER$(G, x)$

---

1 **if** *preprocessing phase **fails*** **then**

2      return **fail**

3 **if** SEARCHINDEX$(H, \ell, x)$ *return **outlier*** **then**

4      return a random index$\in [k]$

5 **else**

6      return SEARCHINDEX$(H, \ell, x)$

---

### 3.3 Analysis of the oracle

We now prove the following property: for most vertex pairs $x, y$, if $x, y$ are in the same cluster, then $\langle f_x, f_y \rangle$ is rough $O(\frac{k}{n})$ (Lemma 3.4); and if $x, y$ are in the different clusters, then $\langle f_x, f_y \rangle$ is close to 0 (Lemma 3.5). We make use of the following three lemmas. Due to the limited space, all the missing proofs will be given in Appendix D.

The following lemma shows that for most vertices $x$, the norm $\|f_x\|_2$ is small.

**Lemma 3.1.** *Let $\alpha \in (0,1)$. Let $k \geq 2$ be an integer, $\varphi \in (0,1)$, and $\varepsilon \in (0,1)$. Let $G = (V,E)$ be a $d$-regular $(k,\varphi,\varepsilon)$-clusterable graph with $|V| = n$. There exists a subset $\widehat{V} \subseteq V$ with $|\widehat{V}| \geq (1-\alpha)|V|$ such that for all $x \in \widehat{V}$, it holds that $\|f_x\|_2 \leq \sqrt{\frac{1}{\alpha} \cdot \frac{k}{n}}$.*

We then show that for most vertices $x$, $f_x$ is close to its center $\mu_x$ of the cluster containing $x$.

**Lemma 3.2.** *Let $\beta \in (0,1)$. Let $k \geq 2$ be an integer, $\varphi \in (0,1)$, and $\varepsilon \in (0,1)$. Let $G = (V,E)$ be a $d$-regular graph that admits a $(k,\varphi,\varepsilon)$-clustering $C_1, \ldots, C_k$ with $|V| = n$. There exists a subset $\widetilde{V} \subseteq V$ with $|\widetilde{V}| \geq (1-\beta)|V|$ such that for all $x \in \widetilde{V}$, it holds that $\|f_x - \mu_x\|_2 \leq \sqrt{\frac{4k\varepsilon}{\beta\varphi^2} \cdot \frac{1}{n}}$.*

The next lemma shows that for most vertices $x$ in a cluster $C_i$, the inner product $\langle f_x, \mu_i \rangle$ is large.

**Lemma 3.3.** *Let $k \geq 2$ be an integer, $\varphi \in (0,1)$, and $\frac{\varepsilon}{\varphi^2}$ be smaller than a sufficiently small constant. Let $G = (V,E)$ be a $d$-regular graph that admits a $(k,\varphi,\varepsilon)$-clustering $C_1, \ldots, C_k$. Let $C_i$ denote the cluster corresponding to the center $\mu_i$, $i \in [k]$. Then for every $C_i$, $i \in [k]$, there exists a subset $\widetilde{C_i} \subseteq C_i$ with $|\widetilde{C_i}| \geq (1 - \frac{10^4 \varepsilon}{\varphi^2})|C_i|$ such that for all $x \in \widetilde{C_i}$, it holds that $\langle f_x, \mu_i \rangle \geq 0.96\|\mu_i\|_2^2$.*

For the sake of description, we introduce the following definition.

**Definition 3.1** (Good and bad vertices). Let $k \geq 2$ be an integer, $\varphi \in (0,1)$, and $\frac{\varepsilon}{\varphi^2}$ be smaller than a sufficiently small constant. Let $G = (V,E)$ be a $d$-regular $n$-vertex graph that admits a $(k,\varphi,\varepsilon)$-clustering $C_1, \ldots, C_k$. We call a vertex $x \in V$ a *good vertex with respect to* $\alpha \in (0,1)$ *and* $\beta \in (0,1)$ if $x \in (\widehat{V} \cap \widetilde{V} \cap (\cup_{i=1}^{k} \widetilde{C_i}))$, where $\widehat{V}$ is the set as defined in Lemma 3.1, $\widetilde{V}$ is the set as defined in Lemma 3.2 and $\widetilde{C_i}$ ($i \in [k]$) is the set as defined in Lemma 3.3. We call a vertex $x \in V$ a *bad vertex with respect to* $\alpha \in (0,1)$ *and* $\beta \in (0,1)$ if it's not a good vertex with respect to $\alpha$ and $\beta$.

Note that for a good vertex $x$ with respect to $\alpha \in (0,1)$ and $\beta \in (0,1)$, the following hold: (1) $\|f_x\|_2 \leq \sqrt{\frac{1}{\alpha} \cdot \frac{k}{n}}$; (2) $\|f_x - \mu_x\|_2 \leq \sqrt{\frac{4k\varepsilon}{\beta\varphi^2} \cdot \frac{1}{n}}$; (3) $\langle f_x, \mu_x \rangle \geq 0.96\|\mu_x\|_2^2$. For a bad vertex $x$ with respect to $\alpha \in (0,1)$ and $\beta \in (0,1)$, it does not satisfy at least one of the above three conditions.

The following lemma shows that if vertex $x$ and vertex $y$ are in the same cluster and both of them are good vertices with respect to $\alpha$ and $\beta$ ($\alpha$ and $\beta$ should be chosen appropriately), then the spectral dot product $\langle f_x, f_y \rangle$ is roughly $0.96 \cdot \frac{1}{|C_i|}$.

**Lemma 3.4.** *Let $k \geq 2$, $\varphi \in (0,1)$ and $\frac{\varepsilon}{\varphi^2}$ be smaller than a sufficiently small constant. Let $G = (V,E)$ be a $d$-regular $n$-vertex graph that admits a $(k,\varphi,\varepsilon)$-clustering $C_1, \ldots, C_k$. Suppose that $x,y \in V$ are in the same cluster $C_i$, $i \in [k]$ and both of them are good vertices with respect to $\alpha = 2\sqrt{k} \cdot (\frac{\varepsilon}{\varphi^2})^{1/3}$ and $\beta = 2\sqrt{k} \cdot (\frac{\varepsilon}{\varphi^2})^{1/3}$. Then $\langle f_x, f_y \rangle \geq 0.96 \left(1 - \frac{4\sqrt{\varepsilon}}{\varphi}\right) \frac{1}{|C_i|} - \frac{\sqrt{k}}{n} \cdot \left(\frac{\varepsilon}{\varphi^2}\right)^{1/6}$.*

The following lemma shows that if vertex $x$ and vertex $y$ are in different clusters and both of them are good vertices with respect to $\alpha$ and $\beta$ ($\alpha$ and $\beta$ should be chosen appropriately), then the spectral dot product $\langle f_x, f_y \rangle$ is close to 0.

**Lemma 3.5.** *Let $k \geq 2$, $\varphi \in (0,1)$ and $\frac{\varepsilon}{\varphi^2}$ be smaller than a sufficiently small constant. Let $G = (V,E)$ be a $d$-regular $n$-vertex graph that admits a $(k,\varphi,\varepsilon)$-clustering $C_1, \ldots, C_k$. Suppose that $x \in C_i$, $y \in C_j$, $(i,j \in [k], i \neq j)$ and both of them are good vertices with respect to $\alpha = 2\sqrt{k} \cdot (\frac{\varepsilon}{\varphi^2})^{1/3}$ and $\beta = 2\sqrt{k} \cdot (\frac{\varepsilon}{\varphi^2})^{1/3}$, the following holds:*

$$\langle f_x, f_y \rangle \leq \frac{\sqrt{k}}{n} \cdot \left(\frac{\varepsilon}{\varphi^2}\right)^{1/6} + \frac{\sqrt{2}k^{1/4}}{\sqrt{n}} \cdot \left(\frac{\varepsilon}{\varphi^2}\right)^{1/3} \cdot \sqrt{\left(1 + \frac{4\sqrt{\varepsilon}}{\varphi}\right)\frac{1}{|C_j|}} + \frac{8\sqrt{\varepsilon}}{\varphi} \cdot \frac{1}{\sqrt{|C_i| \cdot |C_j|}}.$$

**Proof of Theorem 1.** Now we prove our main result Theorem 1.

*Proof.* Let $s = \frac{10k \log k}{\gamma}$ be the size of sampling set $S$, let $\alpha = \beta = 2\sqrt{k} \cdot (\frac{\varepsilon}{\varphi^2})^{1/3}$. Recall that we call a vertex $x$ a bad vertex, if $x \in (V \backslash \widehat{V}) \cup (V \backslash \widetilde{V}) \cup (V \backslash (\cup_{i=1}^{k} \widetilde{C_i}))$, where $\widehat{V}, \widetilde{V}, \widetilde{C_i}, i \in [k]$ are defined in Lemma 3.1, 3.2, 3.3 respectively. We use $B$ to denote the set of all bad vertices. Then we have $|B| \leq (\alpha + \beta + \frac{10^4 \varepsilon}{\varphi^2}) \cdot n = (4\sqrt{k} \cdot (\frac{\varepsilon}{\varphi^2})^{1/3} + \frac{10^4 \varepsilon}{\varphi^2}) \cdot n$. We let $\kappa \leq 4\sqrt{k} \cdot (\frac{\varepsilon}{\varphi^2})^{1/3} + \frac{10^4 \varepsilon}{\varphi^2}$

be the fraction of $B$ in $V$. Since $\frac{\varepsilon}{\varphi^2} < \frac{\gamma^3}{4^3 \cdot 10^9 \cdot k^{\frac{9}{2}} \cdot \log^3 k}$, we have $\kappa \leq 4\sqrt{k} \cdot \left(\frac{\varepsilon}{\varphi^2}\right)^{1/3} + \frac{10^4 \varepsilon}{\varphi^2} \leq \frac{\gamma}{10^3 k \log k} + \frac{\gamma^3}{4^3 \cdot 10^5 \cdot k^{\frac{9}{2}} \log^3 k} \leq \frac{2\gamma}{10^3 k \log k} = \frac{1}{50s}$.

Therefore, by union bound, with probability at least $1 - \kappa \cdot s \geq 1 - \frac{1}{50s} \cdot s = 1 - \frac{1}{50}$, all the vertices in $S$ are good (we fixed $\alpha = \beta = 2\sqrt{k} \cdot (\frac{\varepsilon}{\varphi^2})^{1/3}$, so we will omit "with respect to $\alpha$ and $\beta$" in the following). In the following, we will assume all the vertices in $S$ are good.

Recall that for $i \in [k]$, $|C_i| \geq \gamma \frac{n}{k}$, so with probability at least $1 - (1 - \frac{\gamma}{k})^s = 1 - (1 - \frac{1}{\frac{k}{\gamma}})^{\frac{k}{\gamma} \cdot 10 \log k} \geq 1 - \frac{1}{k^{10}} \geq 1 - \frac{1}{50k}$, there exists at least one vertex in $S$ that is from $C_i$. Then with probability at least $1 - \frac{1}{50}$, for all $k$ clusters $C_1, \ldots, C_k$, there exists at least one vertex in $S$ that is from $C_i$.

Let $\xi = \frac{\sqrt{\gamma}}{1000}$. By Lemma 2.1, we know that with probability at least $1 - \frac{1}{n^{100}}$, for any pair of $x, y \in V$, SPECTRALDOTPRODUCTORACLE$(G, x, y, 1/2, \xi, \mathcal{D})$ computes an output value $\langle f_x, f_y \rangle_{\text{apx}}$ such that $\left| \langle f_x, f_y \rangle_{\text{apx}} - \langle f_x, f_y \rangle \right| \leq \frac{\xi}{n}$. So, with probability at least $1 - \frac{s \cdot s}{n^{100}} \geq 1 - \frac{1}{n^{50}}$, for all pairs $x, y \in S$, SPECTRALDOTPRODUCTORACLE$(G, x, y, 1/2, \xi, \mathcal{D})$ computes an output value $\langle f_x, f_y \rangle_{\text{apx}}$ such that $\left| \langle f_x, f_y \rangle_{\text{apx}} - \langle f_x, f_y \rangle \right| \leq \frac{\xi}{n}$. In the following, we will assume the above inequality holds for any $x, y \in S$.

By Lemma 3.4, we know that if $x, y$ are in the same cluster and both of them are good vertices, then we have $\langle f_x, f_y \rangle \geq 0.96(1 - \frac{4\sqrt{\varepsilon}}{\varphi}) \frac{1}{|C_i|} - \frac{\sqrt{k}}{n} \cdot (\frac{\varepsilon}{\varphi^2})^{1/6} \geq 0.96(1 - \frac{4\sqrt{\varepsilon}}{\varphi}) \frac{\gamma k}{n} - \frac{\sqrt{k}}{n}(\frac{\varepsilon}{\varphi^2})^{1/6}$ since $|C_i| \leq \frac{n}{\gamma k}$. By Lemma 3.5, we know that if $x, y$ are in the different clusters and both of them are good vertices, then we have $\langle f_x, f_y \rangle \leq \frac{\sqrt{k}}{n} \cdot (\frac{\varepsilon}{\varphi^2})^{1/6} + \frac{\sqrt{2}k^{1/4}}{\sqrt{n}} \cdot (\frac{\varepsilon}{\varphi^2})^{1/3} \cdot \sqrt{(1 + \frac{4\sqrt{\varepsilon}}{\varphi}) \frac{1}{|C_j|}} + \frac{8\sqrt{\varepsilon}}{\varphi} \cdot \frac{1}{\sqrt{|C_i| \cdot |C_j|}} \leq \frac{\sqrt{k}}{n} \cdot (\frac{\varepsilon}{\varphi^2})^{1/6} + \sqrt{\frac{2}{\gamma}} \frac{k^{3/4}}{n} \sqrt{1 + \frac{4\sqrt{\varepsilon}}{\varphi}} \cdot (\frac{\varepsilon}{\varphi^2})^{1/3} + \frac{8\sqrt{\varepsilon}}{\varphi} \cdot \frac{k}{\gamma n}$ since $\frac{\gamma n}{k} \leq |C_i|$ for all $i \in [k]$.

Recall that $\frac{\varepsilon}{\varphi^2} < \frac{\gamma^3}{4^3 \cdot 10^9 \cdot k^{\frac{9}{2}} \cdot \log^3 k}$ and $\gamma \in (0.001, 1]$. Let $\theta = 0.96(1 - \frac{4\sqrt{\varepsilon}}{\varphi}) \frac{\gamma k}{n} - \frac{\sqrt{k}}{n}(\frac{\varepsilon}{\varphi^2})^{1/6} - \frac{\xi}{n}$, then we have $\theta > \frac{\sqrt{\gamma}}{n} \cdot (0.96\sqrt{\gamma}k - \frac{0.48}{10^{9/2} \cdot k^{5/4} \log^{3/2} k} - \frac{1}{2 \cdot 10^{3/2} \cdot k^{1/4} \log^{1/2} k} - \frac{1}{1000}) > 0.034 \cdot \frac{\sqrt{\gamma}}{n}$. Let $S$ satisfies that all the vertices in $S$ are good, and $S$ contains at least one vertex from $C_i$ for all $i = 1, \ldots, k$. For any $x, y \in S$, then:

1. If $x, y$ belong to the same cluster, by above analysis, we know that $\langle f_x, f_y \rangle \geq 0.96(1 - \frac{4\sqrt{\varepsilon}}{\varphi}) \frac{\gamma k}{n} - \frac{\sqrt{k}}{n}(\frac{\varepsilon}{\varphi^2})^{1/6}$. Then it holds that $\langle f_x, f_y \rangle_{\text{apx}} \geq \langle f_x, f_y \rangle - \frac{\xi}{n} \geq 0.96(1 - \frac{4\sqrt{\varepsilon}}{\varphi}) \frac{\gamma k}{n} - \frac{\sqrt{k}}{n}(\frac{\varepsilon}{\varphi^2})^{1/6} - \frac{\xi}{n} = \theta$. Thus, an edge $(x, y)$ will be added to $H$ (at lines 8 and 9 of Alg.1).

2. If $x, y$ belong to two different clusters, by above analysis, we know that $\langle f_x, f_y \rangle \leq \frac{\sqrt{k}}{n} \cdot (\frac{\varepsilon}{\varphi^2})^{1/6} + \sqrt{\frac{2}{\gamma}} \frac{k^{3/4}}{n} \sqrt{1 + \frac{4\sqrt{\varepsilon}}{\varphi}} \cdot (\frac{\varepsilon}{\varphi^2})^{1/3} + \frac{8\sqrt{\varepsilon}}{\varphi} \cdot \frac{k}{\gamma n}$. Then it holds that $\langle f_x, f_y \rangle_{\text{apx}} \leq \langle f_x, f_y \rangle + \frac{\xi}{n} \leq \frac{\sqrt{k}}{n} \cdot (\frac{\varepsilon}{\varphi^2})^{1/6} + \sqrt{\frac{2}{\gamma}} \frac{k^{3/4}}{n} \sqrt{1 + \frac{4\sqrt{\varepsilon}}{\varphi}} \cdot (\frac{\varepsilon}{\varphi^2})^{1/3} + \frac{8\sqrt{\varepsilon}}{\varphi} \cdot \frac{k}{\gamma n} + \frac{\xi}{n} < \frac{\sqrt{\gamma}}{n} \cdot (\frac{1}{2 \cdot 10^{3/2} \cdot k^{1/4} \log^{1/2} k} + \frac{1}{2 \cdot 10^3 \cdot k^{3/4} \log k} + \frac{1}{10^{9/2} \cdot k^{5/4} \log^{3/2} k} + \frac{1}{1000}) < 0.027 \cdot \frac{\sqrt{\gamma}}{n} < \theta$, since $\frac{\varepsilon}{\varphi^2} < \frac{\gamma^3}{4^3 \cdot 10^9 \cdot k^{\frac{9}{2}} \cdot \log^3 k}$ and $\xi = \frac{\sqrt{\gamma}}{1000}$. Thus, an edge $(u, v)$ will not be added to $H$.

Therefore, with probability at least $1 - \frac{1}{50} - \frac{1}{50} - \frac{1}{n^{50}} \geq 0.95$, the similarity graph $H$ has following properties: (1) all vertices in $V(H)$ (i.e., $S$) are good; (2) all vertices in $S$ that belongs to the same cluster $C_i$ form a connected components, denoted by $S_i$; (3) there is no edge between $S_i$ and $S_j$, $i \neq j$; (4) there are exactly $k$ connected components in $H$, each corresponding to a cluster.

Now we are ready to consider a query WHICHCLUSTER$(G, x)$.

Assume $x$ is good. We use $C_x$ to denote the cluster that $x$ belongs to. Since all the vertices in $S$ are good, let $y \in C_x \cap S$, so with probability at least $1 - \frac{s}{n^{100}} \geq 1 - \frac{1}{n^{50}}$, by above analysis, we have $\langle f_x, f_y \rangle_{\text{apx}} \geq \langle f_x, f_y \rangle - \frac{\xi}{n} \geq \theta$. On the other hand, for any $y \in S \backslash C_x$, with probability at least $1 - \frac{s}{n^{100}} \geq 1 - \frac{1}{n^{50}}$, by above analysis, we have $\langle f_x, f_y \rangle_{\text{apx}} \leq \langle f_x, f_y \rangle + \frac{\xi}{n} < \theta$. Thus, WHICHCLUSTER$(G, x)$ will output the label of $y \in C_x \cap S$ as $x's$ label (at line 3 of Alg.2).

Therefore, with probability at least $1 - \frac{1}{50} - \frac{1}{50} - \frac{1}{n^{50}} - \frac{n}{n^{50}} \geq 0.95$, all the good vertices will be correctly recovered. So the misclassified vertices come from $B$. We know that $|B| \leq \left(\alpha + \beta + \frac{10^4 \varepsilon}{\varphi^2}\right) \cdot n = \left(4\sqrt{k} \cdot \left(\frac{\varepsilon}{\varphi^2}\right)^{1/3} + \frac{10^4 \varepsilon}{\varphi^2}\right) \cdot n$. Since $|C_i| \geq \frac{\gamma n}{k}$, we have $n \leq \frac{k}{\gamma}|C_i|$. So, $|B| \leq (4\sqrt{k} \cdot (\frac{\varepsilon}{\varphi^2})^{1/3} + \frac{10^4 \varepsilon}{\varphi^2}) \cdot \frac{k}{\gamma}|C_i| \leq O(\frac{k^{\frac{3}{2}}}{\gamma} \cdot (\frac{\varepsilon}{\varphi^2})^{1/3})|C_i|$. This implies that there exists a permutation $\pi : [k] \to [k]$ such that for all $i \in [k]$, $|U_{\pi(i)} \triangle C_i| \leq O(\frac{k^{\frac{3}{2}}}{\gamma} \cdot (\frac{\varepsilon}{\varphi^2})^{1/3})|C_i|$.

**Running time.** By Lemma 2.1, we know that INITIALIZEORACLE$(G, 1/2, \xi)$ computes in time $(\frac{k}{\xi})^{O(1)} \cdot n^{1/2 + O(\varepsilon/\varphi^2)} \cdot (\log n)^3 \cdot \frac{1}{\varphi^2}$ a sublinear space data structure $\mathcal{D}$ and for every pair of vertices $x, y \in V$, SPECTRALDOTPRODUCTORACLE$(G, x, y, 1/2, \xi, \mathcal{D})$ computes an output value $\langle f_x, f_y \rangle_{\text{apx}}$ in $(\frac{k}{\xi})^{O(1)} \cdot n^{1/2 + O(\varepsilon/\varphi^2)} \cdot (\log n)^2 \cdot \frac{1}{\varphi^2}$ time.

In preprocessing phase, for algorithm CONSTRUCTORACLE$(G, k, \varphi, \varepsilon, \gamma)$, it invokes INITIALIZEORACLE one time to construct a data structure $\mathcal{D}$ and SPECTRALDOTPRODUCTORACLE $s \cdot s$ times to construct a similarity graph $H$. So the preprocessing time of our oracle is $(\frac{k}{\xi})^{O(1)} \cdot n^{1/2 + O(\varepsilon/\varphi^2)} \cdot (\log n)^3 \cdot \frac{1}{\varphi^2} + \frac{100k^2 \log^2 k}{\gamma^2} \cdot (\frac{k}{\xi})^{O(1)} \cdot n^{1/2 + O(\varepsilon/\varphi^2)} \cdot (\log n)^2 \cdot \frac{1}{\varphi^2} = O(n^{1/2 + O(\varepsilon/\varphi^2)} \cdot \text{poly}(\frac{k \cdot \log n}{\gamma \varphi}))$.

In query phase, to answer the cluster index that $x$ belongs to, algorithm WHICHCLUSTER$(G, x)$ invokes SPECTRALDOTPRODUCTORACLE $s$ times. So the query time of our oracle is $\frac{10k \log k}{\gamma} \cdot (\frac{k}{\xi})^{O(1)} \cdot n^{1/2 + O(\varepsilon/\varphi^2)} \cdot (\log n)^2 \cdot \frac{1}{\varphi^2} = O(n^{1/2 + O(\varepsilon/\varphi^2)} \cdot \text{poly}(\frac{k \cdot \log n}{\gamma \varphi}))$.

**Space.** By the proof of Lemma 2.1 in [14], we know that overall both algorithm INITIALIZEORACLE and SPECTRALDOTPRODUCTORACLE take $(\frac{k}{\xi})^{O(1)} \cdot n^{1/2 + O(\varepsilon/\varphi^2)} \cdot \text{poly}(\log n)$ space. Therefore, overall preprocessing phase takes $(\frac{k}{\xi})^{O(1)} \cdot n^{1/2 + O(\varepsilon/\varphi^2)} \cdot \text{poly}(\log n) = O(n^{1/2 + O(\varepsilon/\varphi^2)} \cdot \text{poly}(\frac{k \log n}{\gamma}))$ space (at lines 5 and 7 of Alg.1). In query phase, WHICHCLUSTER query takes $(\frac{k}{\xi})^{O(1)} \cdot n^{1/2 + O(\varepsilon/\varphi^2)} \cdot \text{poly}(\log n) = O(n^{1/2 + O(\varepsilon/\varphi^2)} \cdot \text{poly}(\frac{k \log n}{\gamma}))$ space (at line 2 of Alg.2). □

## 4 Experiments

To evaluate the performance of our oracle, we conducted experiments on the random graph generated by the Stochastic Block Model (SBM). In this model, we are given parameters $p, q$ and $n, k$, where $n, k$ denote the number of vertices and the number of clusters respectively; $p$ denotes the probability that any pair of vertices within each cluster is connected by an edge, and $q$ denotes the probability that any pair of vertices from different clusters is connected by an edge. Setting $\frac{p}{q} > c$ for some big enough constant $c$ we can get a well-clusterable graph. All experiments were implemented in Python 3.9 and the experiments were performed using an Intel(R) Core(TM) i7-12700F CPU @ 2.10GHz processor, with 32 GB RAM. Due to the limited space, practical changes to our oracle will be shown in Appendix F.

**Misclassification error.** To evaluate the misclassification error our oracle, we conducted this experiment. In this experiment, we set $k = 3$, $n = 3000$, $q = 0.002$, $p \in [0.02, 0.07]$ in the SBM, where each cluster has 1000 vertices. For each graph $G = (V, E)$, we run WHICHCLUSTER$(G, x)$ for all $x \in V$ and get a $k$-partition $U_1, \ldots, U_k$ of $V$. In experiments, the *misclassification error* of our oracle is defined to be $1 - \frac{1}{n} \cdot \max_\pi \sum_{i=1}^k U_{\pi(i)} \cap C_i$, where $\pi : [k] \to [k]$ is a permutation and $C_1, \ldots, C_k$ are the ground-truth clusters of $G$.

Moreover, we implemented the oracle in [8] to compare with our oracle[2]. The oracle in [8] can be seen as a non-robust version of oracle in [30]. Note that our primary advancement over [8] (also [30]) is evident in the significantly reduced conductance gap we achieve.

---

[2]We remark that the oracle is implicit in [8] (see also [30]). Instead of using the inner product of spectral embeddings of vertex pairs, the authors of [8] used the pairwise $\ell_2$-distance between the distributions of two random walks starting from the two corresponding vertices.

We did not compare with the oracle in [14], since implementing the oracle in [14] poses challenges. As described in section 3.1, the oracle in [14] initially approximates the $k$ cluster centers by sampling around $O(1/\varepsilon \cdot k^4 \log k)$ vertices, and subsequently undertakes the enumeration of approximately $2^{O(1/\varepsilon \cdot k^4 \log^2 k)}$ potential $k$-partitions (Algorithm 10 in [14]). This enumeration process is extremely time-intensive and becomes impractical even for modest values of $k$.

According to the result of our experiment (Table 1), the misclassification error of our oracle is reported to be quite small when $p \geq 0.025$, and even decreases to 0 when $p \geq 0.035$. The outcomes of our experimentation distinctly demonstrate that our oracle's misclassification error remains notably minimal in instances where the input graph showcases an underlying latent cluster structure. In addition, Table 1 also shows that our oracle can handle graphs with a smaller conductance gap than the oracle in [8], which is consistent with the theoretical results. This empirical validation reinforces the practical utility and efficacy of our oracle beyond theoretical conjecture.

Table 1: The misclassification error of the oracle in [8] and our oracle

| $p$ | 0.02 | 0.025 | 0.03 | 0.035 | 0.04 | 0.05 | 0.06 | 0.07 |
|---|---|---|---|---|---|---|---|---|
| min-error[1]([8]) | - | 0.5570 | 0.1677 | 0.0363 | 0.0173 | 0.0010 | 0 | 0 |
| max-error[1]([8]) | - | 0.6607 | 0.6610 | 0.6533 | 0.4510 | 0.0773 | 0.0227 | 0.0013 |
| average-error[1]([8]) | - | 0.6208 | 0.4970 | 0.1996 | 0.0829 | 0.0168 | 0.0030 | 0.0003 |
| error (our) | 0.2273 | 0.0113 | 0.0003 | 0 | 0 | 0 | 0 | 0 |

[1] In the experiment, we found that the misclassification error of oracle in [8] fluctuated greatly, so for the oracle in [8], for each value of $p$, we conducted 30 independent experiments and recorded the minimum error, maximum error and average error of oracle in [8].

**Robustness experiment.** The base graph $G_0 = (V, E)$ is generated from SBM with $n = 3000, k = 3, p = 0.05, q = 0.002$. Note that randomly deleting some edges in each cluster is equivalent to reducing $p$ and randomly deleting some edges between different clusters is equivalent to reducing $q$. So we consider the worst case. We randomly choose one vertex from each cluster; for each selected vertex $x_i$, we randomly delete delNum edges connected to $x_i$ in cluster $C_i$. If $x_i$ has fewer than delNum neighbors within $C_i$, then we delete all the edges incident to $x_i$ in $C_i$. We run WHICHCLUSTER queries for all vertices in $V$ on the resulting graph. We repeated this process for five times for each parameter delNum and recorded the average misclassification error (Table 2). The results show that our oracle is robust against a few number of random edge deletions.

Table 2: The average misclassification error with respect to delNum random edge deletions

| delNum | 25 | 32 | 40 | 45 | 50 | 55 | 60 | 65 |
|---|---|---|---|---|---|---|---|---|
| error | 0.00007 | 0.00007 | 0.00013 | 0.00047 | 0.00080 | 0.00080 | 0.00080 | 0.00087 |

We also conducted experiments to report the query complexity and running time of our oracle. See Appendix F for more details.

## 5 Conclusion

We have devised a new spectral clustering oracle with sublinear preprocessing and query time. In comparison to the approach presented in [14], our oracle exhibits improved preprocessing efficiency, albeit with a slightly higher misclassification error rate. Furthermore, our oracle can be readily implemented in practical settings, while the clustering oracle proposed in [14] poses challenges in terms of implementation feasibility. To obtain our oracle, we have established a property regarding the spectral embeddings of the vertices in $V$ for a $d$-bounded $n$-vertex graph $G = (V, E)$ that exhibits a $(k, \varphi, \varepsilon)$-clustering $C_1, \ldots, C_k$. Specifically, if $x$ and $y$ belong to the same cluster, the dot product of their spectral embeddings (denoted as $\langle f_x, f_y \rangle$) is approximately $O(\frac{k}{n})$. Conversely, if $x$ and $y$ are from different clusters, $\langle f_x, f_y \rangle$ is close to 0. We also show that our clustering oracle is robust against a few random edge deletions and conducted experiments on synthetic networks to validate our theoretical results.

## Acknowledgments and Disclosure of Funding

The work is supported in part by the Huawei-USTC Joint Innovation Project on Fundamental System Software, NSFC grant 62272431 and "the Fundamental Research Funds for the Central Universities".

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

# Appendix

## A  Complete preliminaries

Let $G = (V, E)$ denote a $d$-regular undirected and unweighted graph, where $V := \{1, \ldots, n\}$. Throughout the paper, we use $i \in [n]$ to denote $1 \le i \le n$ and all the vectors will be column vectors unless otherwise specified or transposed to row vectors. For a vertex $x \in V$, let $\mathbb{1}_x \in \mathbb{R}^n$ denote the indicator of $x$, which means $\mathbb{1}_x(i) = 1$ if $i = x$ and $0$ otherwise. For a vector $\mathbf{x}$, we let $\|\mathbf{x}\|_2 = \sqrt{\sum_i \mathbf{x}(i)^2}$ denote its $\ell_2$ norm. For a matrix $A \in \mathbb{R}^{n \times n}$, we use $\|A\|$ to denote the spectral norm of $A$, and we use $\|A\|_F$ to denote the Frobenius norm of $A$. For any two vectors $\mathbf{x}, \mathbf{y} \in \mathbb{R}^n$, we let $\langle \mathbf{x}, \mathbf{y} \rangle = \mathbf{x}^T \mathbf{y}$ denote the dot product of $\mathbf{x}$ and $\mathbf{y}$. For a matrix $A \in \mathbb{R}^{n \times n}$, we use $A_{[i]} \in \mathbb{R}^{n \times i}$ to denote the first $i$ columns of $A$, $1 \le i \le n$.

Let $A \in \mathbb{R}^{n \times n}$ denote the adjacency matrix of $G$ and let $D \in \mathbb{R}^{n \times n}$ denote a diagonal matrix. For the adjacency matrix $A$, $A(i, j) = 1$ if $(i, j) \in E$ and $0$ otherwise, $u, v \in [n]$. For the diagonal matrix $D$, $D(i, i) = \deg(i)$, where $\deg(i)$ is the degree of vertex $i, i \in [n]$. We denote with $L$ the normalized Laplacian of $G$ where $L = D^{-1/2}(D - A)D^{-1/2} = I - \frac{A}{d}$. For $L$, we use $0 \le \lambda_1 \le \cdots \le \lambda_n \le 2$ [7] to denote its eigenvalues and we use $u_1, \ldots, u_n \in \mathbb{R}^n$ to denote the corresponding eigenvectors. Note that the corresponding eigenvectors are not unique, in this paper, we let $u_1, \ldots, u_n$ be an orthonormal basis of eigenvectors of $L$. Let $U \in \mathbb{R}^{n \times n}$ be a matrix whose $i$-th column is $u_i, i \in [n]$, then for every vertex $x \in V$, $f_x = U_{[k]}^T \mathbb{1}_x$. For any two sets $S_1$ and $S_2$, we let $S_1 \triangle S_2$ denote the symmetric difference between $S_1$ and $S_2$.

## B  $d$-bounded graphs to $d$-regular graphs

For a $d$-bounded graph $G' = (V, E)$, we can get a $d$-regular graph $G$ from $G'$ by adding $d - \deg(x)$ self-loops with weight $1/2$ to each vertex $x \in V$. Note that the lazy random walk on $G$ is equivalent to the random walk on $G'$, with the random walk satisfying that if we are at vertex $x$, then we jump to a random neighbor with probability $\frac{1}{2d}$ and stay at $x$ with probability $1 - \frac{\deg(x)}{2d}$. We use $w_{self}(x) = (d - \deg(x)) \cdot \frac{1}{2}$ to denote the the weight of all self-loops of $x \in V$.

## C  Formal statement of Lemma 2.1 and description of dot-product algorithms in [14]

**Lemma C.1** (Lemma 2.1, Formal; Theorem 2, [14]). *Let $\varepsilon, \varphi \in (0, 1)$ with $\varepsilon \le \frac{\varphi^2}{10^5}$. Let $G = (V, E)$ be a $d$-regular graph that admits a $(k, \varphi, \varepsilon)$-clustering $C_1, \ldots, C_k$. Let $\frac{1}{n^5} < \xi < 1$. Then* INITIALIZEORACLE$(G, 1/2, \xi)$ *computes in time* $(\frac{k}{\xi})^{O(1)} \cdot n^{1/2 + O(\varepsilon/\varphi^2)} \cdot (\log n)^3 \cdot \frac{1}{\varphi^2}$ *a sublinear space data structure $\mathcal{D}$ of size* $(\frac{k}{\xi})^{O(1)} \cdot n^{1/2 + O(\varepsilon/\varphi^2)} \cdot (\log n)^3$ *such that with probability at least $1 - n^{-100}$ the following property is satisfied:*

*For every pair of vertices $x, y \in V$,* SPECTRALDOTPRODUCT$(G, x, y, 1/2, \xi, \mathcal{D})$ *computes an output value $\langle f_x, f_y \rangle_{\mathrm{apx}}$ such that with probability at least $1 - n^{-100}$*

$$\left| \langle f_x, f_y \rangle_{\mathrm{apx}} - \langle f_x, f_y \rangle \right| \le \frac{\xi}{n}.$$

*The running time of* SPECTRALDOTPRODUCT$(G, x, y, 1/2, \xi, \mathcal{D})$ *is* $(\frac{k}{\xi})^{O(1)} \cdot n^{1/2 + O(\varepsilon/\varphi^2)} \cdot (\log n)^2 \cdot \frac{1}{\varphi^2}$.

---

**Algorithm 4:** RUNRANDOMWALKS$(G, R, t, x)$

---
**1** Run $R$ random walks of length $t$ starting from $x$
**2** Let $\widehat{m}_x(y)$ be the fraction of random walks that ends at $y$
**3** return $\widehat{m}_x$

---

---

**Algorithm 5:** ESTIMATETRANSITIONMATRIX$(G, I_S, R, t)$

---

**1** **for** *each sample* $x \in I_S$ **do**
**2** $\quad$ $\widehat{m}_x :=$ RUNRANDOMWALKS$(G, R, t, x)$
**3** **end**
**4** Let $\widehat{Q}$ be the matrix whose columns are $\widehat{m}_x$ for $x \in I_S$
**5** **return** $\widehat{Q}$

---

---

**Algorithm 6:** ESTIMATECOLLISIONPROBABILITIES$(G, I_S, R, t)$

---

**1** **for** $i = 1$ *to* $O(\log n)$ **do**
**2** $\quad$ $\widehat{Q}_i :=$ ESTIMATETRANSITIONMATRIX$(G, I_S, R, t)$
**3** $\quad$ $\widehat{P}_i :=$ ESTIMATETRANSITIONMATRIX$(G, I_S, R, t)$
**4** $\quad$ $\mathcal{G}_i := \frac{1}{2}(\widehat{P}_i^T \widehat{Q}_i + \widehat{Q}_i^T \widehat{P}_i)$
**5** **end**
**6** Let $\mathcal{G}$ be a matrix obtained by taking the entrywise median of $\mathcal{G}'_i s$
**7** **return** $\mathcal{G}$

---

---

**Algorithm 7:** INITIALIZEORACLE$(G, \delta, \xi)$ $\quad$ Need: $\varepsilon/\varphi^2 \leq \frac{1}{10^5}$

---

**1** $t := \frac{20 \cdot \log n}{\varphi^2}$
**2** $R_{\text{init}} := O(n^{1-\delta+980 \cdot \varepsilon/\varphi^2} \cdot k^{17}/\xi^2)$
**3** $s := O(n^{480 \cdot \varepsilon/\varphi^2} \cdot \log n \cdot k^8/\xi^2)$
**4** Let $I_S$ be the multiset of $s$ indices chosen independently and uniformly at random from $\{1, \dots, n\}$
**5** **for** $i = 1$ *to* $O(\log n)$ **do**
**6** $\quad$ $\widehat{Q}_i :=$ ESTIMATETRANSITIONMATRIX$(G, I_S, R_{\text{init}}, t)$
**7** **end**
**8** $\mathcal{G} :=$ ESTIMATECOLLISIONPROBABILITIES$(G, I_S, R_{\text{init}}, t)$
**9** Let $\frac{n}{s} \cdot \mathcal{G} := \widehat{W} \widehat{\Sigma} \widehat{W}^T$ be the eigendecomposition of $\frac{n}{s} \cdot \mathcal{G}$
**10** **if** $\widehat{\Sigma}^{-1}$ *exists* **then**
**11** $\quad$ $\Psi := \frac{n}{s} \cdot \widehat{W}_{[k]} \widehat{\Sigma}_{[k]}^{-2} \widehat{W}_{[k]}^T$
**12** $\quad$ **return** $\mathcal{D} := \{\Psi, \widehat{Q}_1, \dots, \widehat{Q}_{O(\log n)}\}$
**13** **end**

---

---

**Algorithm 8:** SPECTRALDOTPRODUCTORACLE$(G, x, y, \delta, \xi, \mathcal{D})$ $\quad$ Need: $\varepsilon/\varphi^2 \leq \frac{1}{10^5}$

---

**1** $R_{\text{query}} := O(n^{\delta+500 \cdot \varepsilon/\varphi^2} \cdot k^9/\xi^2)$
**2** **for** $i = 1$ *to* $O(\log n)$ **do**
**3** $\quad$ $\widehat{m}_x^i :=$ RUNRANDOMWALKS$(G, R_{\text{query}}, t, x)$
**4** $\quad$ $\widehat{m}_y^i :=$ RUNRANDOMWALKS$(G, R_{\text{query}}, t, y)$
**5** **end**
**6** Let $\alpha_x$ be a vector obtained by taking the entrywise median of $(\widehat{Q}_i)^T (\widehat{m}_x^i)$ over all runs
**7** Let $\alpha_y$ be a vector obtained by taking the entrywise median of $(\widehat{Q}_i)^T (\widehat{m}_y^i)$ over all runs
**8** **return** $\langle f_x, f_y \rangle_{\text{apx}} := \alpha_x^T \Psi \alpha_y$

---

## D  Deferred proofs

**Lemma D.1** (Restatement of Lemma 3.1). *Let $\alpha \in (0, 1)$. Let $k \geq 2$ be an integer, $\varphi \in (0, 1)$, and $\varepsilon \in (0, 1)$. Let $G = (V, E)$ be a d-regular $(k, \varphi, \varepsilon)$-clusterable graph with $|V| = n$. There exists a subset $\widehat{V} \subseteq V$ with $|\widehat{V}| \geq (1 - \alpha)|V|$ such that for all $x \in \widehat{V}$, it holds that $\|f_x\|_2 \leq \sqrt{\frac{1}{\alpha} \cdot \frac{k}{n}}$.*

*Proof.* Recall that $u_1, \ldots, u_k$ are an orthonormal basis of eigenvectors of $L$, so $\|u_i\|_2^2 = 1$ for all $i \in [k]$. So $\sum_{i=1}^k \|u_i\|_2^2 = \sum_{i=1}^n \|f_{x_i}\|_2^2 = k$. Let $X$ be a random variable such that $X = \|f_{x_i}\|_2^2$ with probability $\frac{1}{n}$, for each $i \in [n]$. Then we have $\mathrm{E}[X] = \frac{1}{n}\sum_{i=1}^n \|f_{x_i}\|_2^2 = \frac{k}{n}$. Using Markov's inequality, we have $\Pr[X \geq \frac{1}{\alpha} \cdot \mathrm{E}[X]] = \Pr[X \geq \frac{1}{\alpha} \cdot \frac{k}{n}] \leq \alpha$. This gives us that $\Pr[X \leq \frac{1}{\alpha} \cdot \frac{k}{n}] \geq 1 - \alpha$, which means that at least $(1 - \alpha)$ fraction of vertices in V satisfies $\|f_x\|_2^2 \leq \frac{1}{\alpha} \cdot \frac{k}{n}$. We define $\widehat{V} := \{x \in V : \|f_x\|_2^2 \leq \frac{1}{\alpha} \cdot \frac{k}{n}\}$, then we have $|\widehat{V}| \geq (1 - \alpha)|V|$. This ends the proof. $\square$

**Lemma D.2** (Restatement of Lemma 3.2). *Let $\beta \in (0,1)$. Let $k \geq 2$ be an integer, $\varphi \in (0,1)$, and $\varepsilon \in (0,1)$. Let $G = (V, E)$ be a $d$-regular graph that admits a $(k, \varphi, \varepsilon)$-clustering $C_1, \ldots, C_k$ with $|V| = n$. There exists a subset $\widetilde{V} \subseteq V$ with $|\widetilde{V}| \geq (1 - \beta)|V|$ such that for all $x \in \widetilde{V}$, it holds that $\|f_x - \mu_x\|_2 \leq \sqrt{\frac{4k\varepsilon}{\beta\varphi^2} \cdot \frac{1}{n}}$.*

The following result will be used in our proof:

**Lemma D.3** (Lemma 6, [14]). *Let $k \geq 2$ be an integer, $\varphi \in (0,1)$, and $\varepsilon \in (0,1)$. Let $G = (V, E)$ be a $d$-regular graph that admits a $(k, \varphi, \varepsilon)$-clustering $C_1, \ldots, C_k$. Then for all $\alpha \in \mathbb{R}^k$, with $\|\alpha\|_2 = 1$ we have*

$$\sum_{i=1}^k \sum_{x \in C_i} \langle f_x - \mu_i, \alpha \rangle^2 \leq \frac{4\varepsilon}{\varphi^2}.$$

*Proof of Lemma 3.2.* By summing over $\alpha$ in an orthonormal basis of $\mathbb{R}^k$, we can get

$$\sum_{x \in V} \|f_x - \mu_x\|_2^2 \leq k \cdot \frac{4\varepsilon}{\varphi^2} = \frac{4k\varepsilon}{\varphi^2},$$

where $\mu_x$ is the cluster center of the cluster that $x$ belongs to. Define $V^* = \{x \in V : \|f_x - \mu_x\|_2^2 \geq \frac{4k\varepsilon}{\beta\varphi^2} \cdot \frac{1}{n}\}$. Then,

$$\frac{4k\varepsilon}{\varphi^2} \geq \sum_{x \in V} \|f_x - \mu_x\|_2^2 \geq \sum_{x \in V^*} \|f_x - \mu_x\|_2^2 \geq \sum_{x \in V^*} \frac{4k\varepsilon}{\beta\varphi^2} \cdot \frac{1}{n} = |V^*| \cdot \frac{4k\varepsilon}{\beta\varphi^2} \cdot \frac{1}{n}.$$

So, we can get $|V^*| \leq \beta n$. We define $\widetilde{V} = V \setminus V^* = \{x \in V : \|f_x - \mu_x\|_2^2 \leq \frac{4k\varepsilon}{\beta\varphi^2} \cdot \frac{1}{n}\}$. Therefore, we have $|\widetilde{V}| \geq (1 - \beta)n = (1 - \beta)|V|$. This ends the proof. $\square$

**Lemma D.4** (Restatement of Lemma 3.3). *Let $k \geq 2$ be an integer, $\varphi \in (0,1)$, and $\frac{\varepsilon}{\varphi^2}$ be smaller than a sufficiently small constant. Let $G = (V, E)$ be a $d$-regular graph that admits a $(k, \varphi, \varepsilon)$-clustering $C_1, \ldots, C_k$. Let $C_i$ denote the cluster corresponding to the center $\mu_i$, $i \in [k]$. Then for every $C_i$, $i \in [k]$, there exists a subset $\widehat{C_i} \subseteq C_i$ with $|\widehat{C_i}| \geq (1 - \frac{10^4\varepsilon}{\varphi^2})|C_i|$ such that for all $x \in \widehat{C_i}$, it holds that $\langle f_x, \mu_i \rangle \geq 0.96\|\mu_i\|_2^2$.*

The following result will be used in our proof:

**Lemma D.5** (Lemma 31, [14]). *Let $k \geq 2$, $\varphi \in (0,1)$, and $\frac{\varepsilon}{\varphi^2}$ be smaller than a sufficiently small constant. Let $G = (V, E)$ be a $d$-regular graph that admits a $(k, \varphi, \varepsilon)$-clustering $C_1, \ldots, C_k$. If $\mu_i's$ are cluster means then the following conditions hold. Let $S \subset \{\mu_1, \ldots, \mu_k\}$. Let $\Pi$ denote the orthogonal projection matrix on to the $\mathrm{span}(S)^\perp$. Let $\mu \in \{\mu_1, \ldots, \mu_k\} \setminus S$. Let $C$ denote the cluster corresponding to the center $\mu$. Let*

$$\widehat{C} := \{x \in V : \langle \Pi f_x, \Pi\mu \rangle \geq 0.96\|\Pi\mu\|_2^2\}$$

*then we have:*

$$|C \setminus \widehat{C}| \leq \frac{10^4\varepsilon}{\varphi^2}|C|.$$

*Proof of Lemma 3.3.* We apply $S = \emptyset$ in Lemma D.5 so that $\Pi$ is an identity matrix and we will have $|C_i \setminus \widehat{C_i}| \leq \frac{10^4\varepsilon}{\varphi^2}|C_i|$, where $\widehat{C_i} := \{x \in V : \langle f_x, \mu_i \rangle \geq 0.96\|\mu_i\|_2^2\}$, $i \in [k]$. So

$$|C_i \cap \widehat{C_i}| \geq \left(1 - \frac{10^4\varepsilon}{\varphi^2}\right)|C_i|.$$

We define $\widetilde{C_i} = C_i \cap \widehat{C_i}$, $i \in [k]$. Therefore, for every $C_i$, $i \in [k]$, there exists a subset $\widetilde{C_i} \subseteq C_i$ with $|\widetilde{C_i}| \geq (1 - \frac{10^4 \varepsilon}{\varphi^2})|C_i|$ such that for all $x \in \widetilde{C_i}$, it holds that $\langle f_x, \mu_i \rangle \geq 0.96\|\mu_i\|_2^2$. $\qquad\square$

**Lemma D.6** (Restatement of Lemma 3.4). *Let $k \geq 2$, $\varphi \in (0,1)$ and $\frac{\varepsilon}{\varphi^2}$ be smaller than a sufficiently small constant. Let $G = (V, E)$ be a d-regular n-vertex graph that admits a $(k, \varphi, \varepsilon)$-clustering $C_1, \ldots, C_k$. Suppose that $x, y \in V$ are in the same cluster $C_i, i \in [k]$ and both of them are good vertex with respect to $\alpha = 2\sqrt{k} \cdot (\frac{\varepsilon}{\varphi^2})^{1/3}$ and $\beta = 2\sqrt{k} \cdot (\frac{\varepsilon}{\varphi^2})^{1/3}$, the following holds:*

$$\langle f_x, f_y \rangle \geq 0.96 \left(1 - \frac{4\sqrt{\varepsilon}}{\varphi}\right) \frac{1}{|C_i|} - \frac{\sqrt{k}}{n} \cdot \left(\frac{\varepsilon}{\varphi^2}\right)^{1/6}.$$

The following result will be used in our proof:

**Lemma D.7** (Lemma 7, [14]). *Let $k \geq 2$ be an integer, $\varphi \in (0,1)$, and $\varepsilon \in (0,1)$. Let $G = (V, E)$ be a d-regular graph that admits a $(k, \varphi, \varepsilon)$-clustering $C_1, \ldots, C_k$. Then we have*

1. *for all $i \in [k]$, $\left| \|\mu_i\|_2^2 - \frac{1}{|C_i|} \right| \leq \frac{4\sqrt{\varepsilon}}{\varphi} \frac{1}{|C_i|}$,*
2. *for all $i \neq j \in [k]$, $|\langle \mu_i, \mu_j \rangle| \leq \frac{8\sqrt{\varepsilon}}{\varphi} \frac{1}{\sqrt{|C_i||C_j|}}$.*

*Proof of Lemma 3.4.* According to the distributive law of dot product, we have

$$\langle f_x, f_y \rangle = \langle f_x, f_y - \mu_i + \mu_i \rangle = \langle f_x, f_y - \mu_i \rangle + \langle f_x, \mu_i \rangle.$$

By using Cauchy-Schwarz Inequality, we have $|\langle f_x, f_y - \mu_i \rangle| \leq \|f_x\|_2 \cdot \|f_y - \mu_i\|_2$. Since $x$ and $y$ are both good vertices with respect to $\alpha = 2\sqrt{k} \cdot (\frac{\varepsilon}{\varphi^2})^{1/3}$ and $\beta = 2\sqrt{k} \cdot (\frac{\varepsilon}{\varphi^2})^{1/3}$, we have

$$|\langle f_x, f_y - \mu_i \rangle| \leq \|f_x\|_2 \cdot \|f_y - \mu_i\|_2 \leq \sqrt{\frac{1}{\alpha} \cdot \frac{k}{n}} \sqrt{\frac{4k\varepsilon}{\beta \varphi^2} \cdot \frac{1}{n}} = \frac{\sqrt{k}}{n} \cdot \left(\frac{\varepsilon}{\varphi^2}\right)^{1/6},$$

which gives us that $\langle f_x, f_y - \mu_i \rangle \geq -\frac{\sqrt{k}}{n} \cdot (\frac{\varepsilon}{\varphi^2})^{1/6}$. Recall that $x$ is a good vertex, we have $\langle f_x, \mu_i \rangle \geq 0.96\|\mu_i\|_2^2$. Hence, it holds that

$$\begin{aligned}
\langle f_x, f_y \rangle &= \langle f_x, f_y - \mu_i \rangle + \langle f_x, \mu_i \rangle \\
&\geq 0.96\|\mu_i\|_2^2 - \frac{\sqrt{k}}{n} \cdot \left(\frac{\varepsilon}{\varphi^2}\right)^{1/6} \\
&\geq 0.96 \left(1 - \frac{4\sqrt{\varepsilon}}{\varphi}\right) \frac{1}{|C_i|} - \frac{\sqrt{k}}{n} \cdot \left(\frac{\varepsilon}{\varphi^2}\right)^{1/6}.
\end{aligned}$$

The last inequality is according to item 1 in Lemma D.7. $\qquad\square$

**Lemma D.8** (Restatement of Lemma 3.5). *Let $k \geq 2$, $\varphi \in (0,1)$ and $\frac{\varepsilon}{\varphi^2}$ be smaller than a sufficiently small constant. Let $G = (V, E)$ be a d-regular n-vertex graph that admits a $(k, \varphi, \varepsilon)$-clustering $C_1, \ldots, C_k$. Suppose that $x \in C_i$, $y \in C_j$, $(i, j \in [k], i \neq j)$ and both of them are good vertex with respect to $\alpha = 2\sqrt{k} \cdot (\frac{\varepsilon}{\varphi^2})^{1/3}$ and $\beta = 2\sqrt{k} \cdot (\frac{\varepsilon}{\varphi^2})^{1/3}$, the following holds:*

$$\langle f_x, f_y \rangle \leq \frac{\sqrt{k}}{n} \cdot \left(\frac{\varepsilon}{\varphi^2}\right)^{1/6} + \frac{\sqrt{2}k^{1/4}}{\sqrt{n}} \cdot \left(\frac{\varepsilon}{\varphi^2}\right)^{1/3} \cdot \sqrt{\left(1 + \frac{4\sqrt{\varepsilon}}{\varphi}\right) \frac{1}{|C_j|}} + \frac{8\sqrt{\varepsilon}}{\varphi} \cdot \frac{1}{\sqrt{|C_i| \cdot |C_j|}}.$$

*Proof.* According to the distributive law of dot product, we have

$$\begin{aligned}
\langle f_x, f_y \rangle &= \langle f_x, f_y - \mu_j + \mu_j \rangle \\
&= \langle f_x, f_y - \mu_j \rangle + \langle f_x, \mu_j \rangle \\
&= \langle f_x, f_y - \mu_j \rangle + \langle f_x - \mu_i + \mu_i, \mu_j \rangle \\
&= \langle f_x, f_y - \mu_j \rangle + \langle f_x - \mu_i, \mu_j \rangle + \langle \mu_i, \mu_j \rangle.
\end{aligned}$$

By Cauchy-Schwarz Inequality, we have

$$|\langle f_x, f_y - \mu_j \rangle| \le \|f_x\|_2 \cdot \|f_y - \mu_j\|_2$$

and

$$|\langle f_x - \mu_i, \mu_j \rangle| \le \|\mu_j\|_2 \cdot \|f_x - \mu_i\|_2.$$

Since $x$ and $y$ are both good vertices with respect to $\alpha = 2\sqrt{k} \cdot (\frac{\varepsilon}{\varphi^2})^{1/3}$ and $\beta = 2\sqrt{k} \cdot (\frac{\varepsilon}{\varphi^2})^{1/3}$, we have

$$|\langle f_x, f_y - \mu_j \rangle| \le \|f_x\|_2 \cdot \|f_y - \mu_j\|_2 \le \sqrt{\frac{1}{\alpha} \cdot \frac{k}{n}} \sqrt{\frac{4k\varepsilon}{\beta\varphi^2} \cdot \frac{1}{n}} = \frac{\sqrt{k}}{n} \cdot \left(\frac{\varepsilon}{\varphi^2}\right)^{1/6}$$

and

$$|\langle f_x - \mu_i, \mu_j \rangle| \le \|\mu_j\|_2 \cdot \|f_x - \mu_i\|_2 \le \|\mu_j\|_2 \cdot \sqrt{\frac{4k\varepsilon}{\beta\varphi^2} \cdot \frac{1}{n}} = \frac{\sqrt{2}k^{1/4}}{\sqrt{n}} \cdot \left(\frac{\varepsilon}{\varphi^2}\right)^{1/3} \cdot \|\mu_j\|_2.$$

So we have

$$
\begin{aligned}
\langle f_x, f_y \rangle &= \langle f_x, f_y - \mu_j \rangle + \langle f_x - \mu_i, \mu_j \rangle + \langle \mu_i, \mu_j \rangle \\
&\le \|f_x\|_2 \cdot \|f_y - \mu_j\|_2 + \|\mu_j\|_2 \cdot \|f_x - \mu_i\|_2 + \langle \mu_i, \mu_j \rangle \\
&\le \frac{\sqrt{k}}{n} \cdot \left(\frac{\varepsilon}{\varphi^2}\right)^{1/6} + \frac{\sqrt{2}k^{1/4}}{\sqrt{n}} \cdot \left(\frac{\varepsilon}{\varphi^2}\right)^{1/3} \cdot \|\mu_j\|_2 + \langle \mu_i, \mu_j \rangle \\
&\le \frac{\sqrt{k}}{n} \cdot \left(\frac{\varepsilon}{\varphi^2}\right)^{1/6} + \frac{\sqrt{2}k^{1/4}}{\sqrt{n}} \cdot \left(\frac{\varepsilon}{\varphi^2}\right)^{1/3} \cdot \sqrt{\left(1 + \frac{4\sqrt{\varepsilon}}{\varphi}\right)\frac{1}{|C_j|}} + \frac{8\sqrt{\varepsilon}}{\varphi} \cdot \frac{1}{\sqrt{|C_i| \cdot |C_j|}}.
\end{aligned}
$$

The last inequality is according to item 1 and item 2 in Lemma D.7. $\qquad\square$

# E    Formal statement of Theorem 2 and proof

**Theorem 2** (Formal; Robust against random edge deletions)**.** *Let $k \ge 2$ be an integer, $\varphi \in (0,1)$. Let $G_0 = (V, E_0)$ be a $d$-regular $n$-vertex graph that admits a $(k, \varphi, \varepsilon)$-clustering $C_1, \dots, C_k$, $\frac{\varepsilon}{\varphi^4} \ll \frac{\gamma^3}{k^{\frac{9}{2}} \cdot \log^3 k}$ and for all $i \in [k]$, $\gamma\frac{n}{k} \le |C_i| \le \frac{n}{\gamma k}$, where $\gamma$ is a constant that is in $(0.001, 1]$.*

1. *Let $G$ be a graph obtained from $G_0$ by deleting at most $c$ edges in each cluster, where $c$ is a constant. If $0 \le c \le \frac{d\varphi^2}{2\sqrt{10}}$, then there exists an algorithm that has query access to the adjacency list of $G$ and constructs a clustering oracle in $O(n^{1/2+O(\varepsilon/\varphi^2)} \cdot \text{poly}(\frac{k \log n}{\gamma\varphi}))$ preprocessing time and takes $O(n^{1/2+O(\varepsilon/\varphi^2)} \cdot \text{poly}(\frac{k \log n}{\gamma}))$ space. Furthermore, with probability at least $0.95$, the following hold:*

   1). *Using the oracle, the algorithm can answer any WHICHCLUSTER query in $O(n^{1/2+O(\varepsilon/\varphi^2)} \cdot \text{poly}(\frac{k \log n}{\gamma\varphi}))$ time and a WHICHCLUSTER query takes $O(n^{1/2+O(\varepsilon/\varphi^2)} \cdot \text{poly}(\frac{k \log n}{\gamma}))$ space.*

   2). *Let $U_i := \{x \in V : \text{WHICHCLUSTER}(G, x) = i\}, i \in [k]$ be the clusters recovered by the algorithm. There exists a permutation $\pi : [k] \to [k]$ such that for all $i \in [k]$, $|U_{\pi(i)} \triangle C_i| \le O(\frac{k^{\frac{3}{2}}}{\gamma} \cdot (\frac{\varepsilon}{\varphi^4})^{1/3})|C_i|$.*

2. *Let $G$ be a graph obtained from $G_0$ by randomly deleting at most $O(\frac{kd^2}{\log k + d})$ edges in $G_0$. With probability at least $1 - \frac{1}{k^2}$, then there exists an algorithm that has query access to the adjacency list of $G$ and constructs a clustering oracle in $O(n^{1/2+O(\varepsilon/\varphi^2)} \cdot \text{poly}(\frac{k \log n}{\gamma\varphi}))$ preprocessing time and takes $O(n^{1/2+O(\varepsilon/\varphi^2)} \cdot \text{poly}(\frac{k \log n}{\gamma}))$ space. Furthermore, with probability at least $0.95$, the following hold:*

   1). *Using the oracle, the algorithm can answer any WHICHCLUSTER query in $O(n^{1/2+O(\varepsilon/\varphi^2)} \cdot \text{poly}(\frac{k \log n}{\gamma\varphi}))$ time and a WHICHCLUSTER query takes $O(n^{1/2+O(\varepsilon/\varphi^2)} \cdot \text{poly}(\frac{k \log n}{\gamma}))$ space.*

2). *Let $U_i := \{x \in V : \text{WHICHCLUSTER}(G, x) = i\}, i \in [k]$ be the clusters recovered by the algorithm. There exists a permutation $\pi : [k] \to [k]$ such that for all $i \in [k]$,*

$$|U_{\pi(i)} \triangle C_i| \le O(\tfrac{k^{\frac{3}{2}}}{\gamma} \cdot (\tfrac{\varepsilon}{\varphi^4})^{1/3})|C_i|.$$

To proof Theorem 2, we need the following lemmas.

**Lemma E.1** (Cheeger's Inequality). *In holds for any graph $G$ that*

$$\frac{\lambda_2}{2} \le \phi(G) \le \sqrt{2\lambda_2}.$$

**Lemma E.2** (Weyl's Inequality). *Let $A, B \in \mathbb{R}^{n \times n}$ be symmetric matrices. Let $\alpha_1, \ldots, \alpha_n$ and $\beta_1, \ldots, \beta_n$ be the eigenvalues of $A$ and $B$ respectively. Then for any $i \in [n]$, we have*

$$|\alpha_i - \beta_i| \le \|A - B\|,$$

*where $\|A - B\|$ is the spectral norm of $A - B$.*

*Proof of Theorem 2.* **Proof of item 1:** For any $d$-bounded graph $G' = (V, E)$, we can get a $d$-regular graph $G$ from $G'$ by adding $d - \deg(x)$ self-loops with weight $1/2$ to each vertex $x \in V$. Then according to [7], the normalized Laplacian of $G$ (denoted as $L$) satisfies

$$L(x, y) = \begin{cases} 1 - \frac{w_{self}(x)}{d} & \text{if} \quad x = y \\ -\frac{1}{d} & \text{if} \quad x \ne y \quad \text{and} \quad (x, y) \in E \\ 0 & \text{otherwise} \end{cases}.$$

Let $G_0 = (V, E_0)$ be a $d$-regular graph that admits a $(k, \varphi, \varepsilon)$-clustering $C_1, \ldots, C_k$. Now we consider a cluster $C_i, i \in [k]$. Let $C_i^0$ be a $d$-regular graph obtained by adding $d - deg_i(x)$ self-loops to each vertex $x \in C_i$, where $deg_i(x)$ is the degree of vertex $x$ in the subgraph $C_i$, $i \in [k]$. Let $C_i^j$ be a graph obtained from $C_i^{j-1}$ by: (1) randomly deleting one edge $(u, v) \in E(C_i^{j-1})$, where $E(C_i^{j-1})$ is a set of edges that have both endpoints in $C_i^{j-1}$; (2) turning the result subgraph in (1) be a $d$-regular graph, $i \in [k], j \in [c]$. Let $L_i^j$ be the normalized Laplacian of $C_i^j, i \in [k], j = 0, 1, \ldots, c$. Let $H_i^j = L_i^j - L_i^{j-1}, i \in [k], j \in [c]$. Then if $u \ne v$, we have

$$H_i^j(x, y) = \begin{cases} \frac{1}{d} & \text{if} \quad x = u, y = v \quad \text{or} \quad x = v, y = u \\ -\frac{1}{2d} & \text{if} \quad x = y = u \quad \text{or} \quad x = y = v \\ 0 & \text{otherwise} \end{cases},$$

and if $u = v$, $H_i^j$ is a all-zero matrix. Consider the fact that for a symmetric matrix, the spectral norm is less than or equal to its Frobenius norm, we will have $\|H_i^j\| \le \|H_i^j\|_F = \sqrt{2 \cdot \frac{1}{d^2} + 2 \cdot \frac{1}{4d^2}} = \sqrt{\frac{5}{2d^2}} = \frac{\sqrt{10}}{2d}$ for all $i \in [k], j \in [c]$. Let $H_i = \sum_{j=1}^c H_i^j = L_i^c - L_i^0$, we have $\|H_i\| \le \frac{\sqrt{10}}{2d} \cdot c, i \in [k]$. Let $\lambda_2(L_i^0)$ and $\lambda_2(L_i^c)$ be the second smallest eigenvalue of $L_i^0$ and $L_i^c$ respectively. By Lemma E.2, we have $|\lambda_2(L_i^c) - \lambda_2(L_i^0)| \le \|H_i\| \le \frac{\sqrt{10}}{2d} \cdot c$, which gives us $\lambda_2(L_i^c) \ge \lambda_2(L_i^0) - \frac{\sqrt{10}}{2d} \cdot c, i \in [k]$. By Lemma E.1 and the precondition that $c \le \frac{d\varphi^2}{2\sqrt{10}}$, we have $\lambda_2(L_i^0) \ge \frac{\varphi^2}{2} \ge \frac{\sqrt{10}}{d} \cdot c$. Therefore,

$$\begin{aligned} \lambda_2(L_i^c) &\ge \lambda_2(L_i^0) - \frac{\sqrt{10}}{2d} \cdot c \\ &= \frac{1}{2}\lambda_2(L_i^0) + \frac{1}{2}\lambda_2(L_i^0) - \frac{\sqrt{10}}{2d} \cdot c \\ &\ge \frac{1}{2}\lambda_2(L_i^0) + \frac{\sqrt{10}}{2d} \cdot c - \frac{\sqrt{10}}{2d} \cdot c \\ &= \frac{1}{2}\lambda_2(L_i^0). \end{aligned}$$

Again by Lemma E.1, for graph $C_i^c$, we have $\phi_{\text{in}}(C_i^c) = \phi(C_i^c) \ge \frac{1}{2}\lambda_2(L_i^c) \ge \frac{1}{4}\lambda_2(L_i^0) \ge \frac{1}{8}\varphi^2, i \in [k]$. Note that we slightly abuse the notion $C_i^c$, for $\phi(C_i^c)$, we treat $C_i^c$ as a $d$-regular graph, and

for $\phi_{\text{in}}(C_i^c)$, we treat $C_i^c$ as a cluster obtained by deleting $c$ edges from $E(C_i)$. So, for a $(k, \varphi, \varepsilon)$-clusterable graph $G_0 = (V, E_0)$, if we delete at most $c \leq \frac{d\varphi^2}{2\sqrt{10}}$ edges in each cluster, then the resulting graph $G$ is $(k, \frac{1}{8}\varphi^2, \varepsilon)$-clusterable. Since $\frac{\varepsilon}{\varphi^4} \ll \frac{\gamma^3}{k^{\frac{9}{2}} \cdot \log^3 k}$, we have $\frac{\varepsilon}{(\frac{1}{8}\varphi^2)^2} \ll \frac{\gamma^3}{k^{\frac{9}{2}} \cdot \log^3 k}$. The statement of item 1 in this theorem follows from the same augments as those in the proof of Theorem 1 with parameter $\varphi' = \frac{1}{8}\varphi^2$ in $G$.

**Proof of item 2:** Let $c = \frac{d\varphi^2}{2\sqrt{10}}$. Since $|C_i| \leq \frac{n}{\gamma k}$ for all $i \in [k]$, we have $|E(C_i)| \leq \frac{n}{\gamma k} \cdot \frac{d}{2}$, where $E(C_i)$ is a set of edges that have both endpoints in $C_i$. So $\frac{|E(C_i)|}{|E_0|} \leq \frac{\frac{n}{\gamma k} \cdot \frac{d}{2}}{\frac{nd}{2}} = \frac{1}{\gamma k}, i \in [k]$. Since $|C_i| \geq \frac{\gamma n}{k}$ and $\phi_{\text{out}}(C_i, V) \leq \varepsilon$, we have $E(C_i) \geq \frac{nd}{2k}(\gamma - \frac{\varepsilon}{\gamma})$. So $\frac{|E(C_i)|}{|E_0|} \geq \frac{\frac{nd}{2k}(\gamma - \frac{\varepsilon}{\gamma})}{\frac{nd}{2}} = \frac{1}{k}(\gamma - \frac{\varepsilon}{\gamma}), i \in [k]$. Combining the above two results, we have $\frac{1}{k}(\gamma - \frac{\varepsilon}{\gamma}) \leq \frac{|E(C_i)|}{|E_0|} \leq \frac{1}{\gamma k}$.

In the following, we use $X_i$ to denote the number of edges that are deleted from $E(C_i)$. If we randomly delete $\frac{kc^2\gamma(\gamma^2-\varepsilon)}{9\log k + 2(\gamma^2-\varepsilon)c} = O(\frac{kd^2}{\log k + d})$ edges from $G_0$, then we have $\frac{1}{k}(\gamma - \frac{\varepsilon}{\gamma}) \cdot \frac{kc^2\gamma(\gamma^2-\varepsilon)}{9\log k + 2(\gamma^2-\varepsilon)c} \leq \mathrm{E}(X_i) \leq \frac{1}{\gamma k} \cdot \frac{kc^2\gamma(\gamma^2-\varepsilon)}{9\log k + 2(\gamma^2-\varepsilon)c}$, which gives us

$$\frac{c^2(\gamma^2 - \varepsilon)^2}{9\log k + 2(\gamma^2 - \varepsilon)c} \leq \mathrm{E}(X_i) \leq \frac{c^2(\gamma^2 - \varepsilon)}{9\log k + 2(\gamma^2 - \varepsilon)c}.$$

Chernoff-Hoeffding implies that $\Pr[X_i > (1+\delta)\cdot\mathrm{E}(X_i)] \leq e^{\frac{-\mathrm{E}(X_i)\cdot\delta^2}{3}}$. We set $\delta = \frac{9\log k + 2(\gamma^2-\varepsilon)c}{(\gamma^2-\varepsilon)c} - 1$, then we have

$$\begin{aligned}
\Pr\left[X_i > c\right] = \Pr\left[X_i > (1+\delta) \cdot \frac{c^2(\gamma^2 - \varepsilon)}{9\log k + 2(\gamma^2 - \varepsilon)c}\right] \\
\leq \Pr\left[X_i > (1+\delta) \cdot \mathrm{E}(X_i)\right] \\
\leq e^{\frac{-\mathrm{E}(X_i)\cdot\delta^2}{3}} \\
\leq e^{\frac{-1}{3} \cdot \frac{c^2(\gamma^2-\varepsilon)^2}{9\log k + 2(\gamma^2-\varepsilon)c} \cdot \delta^2} \\
\leq e^{\frac{-1}{3} \cdot \frac{c^2(\gamma^2-\varepsilon)^2}{9\log k + 2(\gamma^2-\varepsilon)c} \cdot (\delta+1)(\delta-1)} \\
= \frac{1}{k^3}.
\end{aligned}$$

Using union bound, with probability at least $1 - \frac{1}{k^3} \cdot k = 1 - \frac{1}{k^2}$, we have that if we randomly delete $\frac{kc^2\gamma(\gamma^2-\varepsilon)}{9\log k + 2(\gamma^2-\varepsilon)c} = O(\frac{kd^2}{\log k + d})$ edges from $G_0$, there is no cluster that is deleted more than $c$ edges. Therefore, according to item 1 of Theorem 2, with probability at least $1 - \frac{1}{k^3} \cdot k = 1 - \frac{1}{k^2}$, $G$ is a $(k, \frac{1}{8}\varphi^2, \varepsilon)$-clusterable graph. The statement of item 2 in this theorem also follows from the same augments as those in the proof of Theorem 1 with parameter $\varphi' = \frac{1}{8}\varphi^2$ in $G$. $\qquad\square$

## F   Experiment Details

**Practical changes to our oracle.** In order to implement our oracle, we need to make some modifications to the theoretical algorithms. To adapt the dot product oracle parameters (see Algorithm 7 and Algorithm 8 in Appendix C), i.e., $t$ (random walk length), $s$ (sampling set size), and $R_{init}$, $R_{query}$ (number of random walks), we exploit the theoretical gap between intra-cluster and inter-cluster dot products in clusterable graphs. Given a clusterable graph $G$, by constructing the dot product oracle with various parameter settings and calculating some intra-cluster and inter-cluster dot products, we generate density graphs. The setting with the most prominent gap in the density graph is selected (see Figure 2 for an illustrative example).

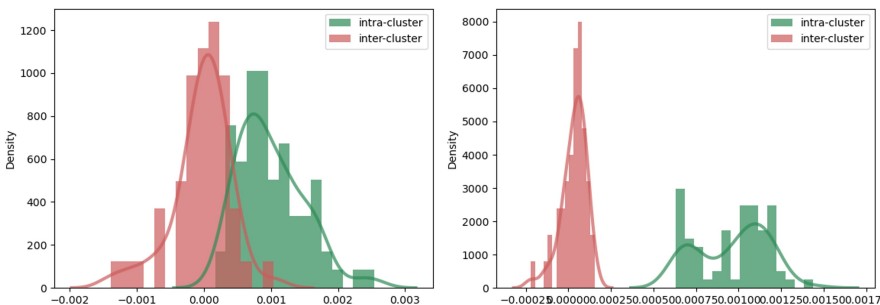

Figure 2: For a random graph $G$ generated by SBM with $n = 3000, k = 3, p = 0.03, q = 0.002$, we build the dot product oracle for several different parameters for $t, s, R_{init}, R_{query}$ and plot the density graph. The setting with the most prominent gap in the density graph, i.e., the one on the right, is selected. We can further set $\theta = 0.0005$ for $G$ according to the right graph.

Determining the appropriate threshold $\theta$ (at lines 2, 8, 9 of Alg.1 and line 3 of Alg.2) is the next step. By observing the density graph linked to the chosen dot product oracle parameters, we identify the fitting $\theta$ (see Figure 2 for an illustrative example).

Determining the appropriate sampling set size $s$ (at line 3 of Alg.1) of our oracle is the final step. Given a graph $G = (V, E)$ generated by SBM, for all vertices in $V$, we know their ground-truth clusters. We can built our clustering oracle for several parameters for $s$. For each parameter setting, we run WHICHCLUSTER$(G, x)$ for some $x \in V$ and check if $x$ was classified correctly. We pick the parameter setting with the most correct answers.

**Query complexity.** We conducted an experiment on a SBM graph with $k = 3, n = 15000, q = 0.002, p = 0.2$. We calculate the fraction of edges that have been accessed given a number of invocations of WHICHCLUSTER$(G, x)$ (Table 3). (Note that there is a trade-off between computational cost and clustering quality. Therefore, it is necessary to point out that the parameters of this experiment are set reasonably and the misclassification error is 0.) Table 3 shows that as long as the number of WHICHCLUSTER queries is not too large, our algorithm only reads a small portion of the input graph.

The above experiment shows that for a small target misclassification error, our algorithms only require a *sublinear amount* of data, which is often critical when analyzing large social networks, since one typically does not have access to the entire network.

Table 3: The fraction of accessed edges of queries

| # queries | 0 | 50 | 100 | 200 | 400 | 800 | 1600 | 3200 |
|---|---|---|---|---|---|---|---|---|
| fraction | 0.1277 | 0.2539 | 0.3637 | 0.5377 | 0.7517 | 0.9273 | 0.9929 | 0.9999 |

**Running time.** To evaluate the running time of our oracle, we conducted this experiment on some random graphs generated by SBM with $n = 3000, k = 3, q = 0.002$ and $p \in [0.02, 0.06]$. Note that there is a trade-off between running time and clustering quality. In this experiment, we set the experimental parameters the same as those in the misclassification error experiment, which can ensure a small error. We recorded the running time of constructing a similarity graph $H$ as construct-time. For each $p$, we query all the vertices in the input graph and recorded the average time of the $n = 3000$ queries as query-time (Table 4).

Table 4: The average query time of our oracle

| $p$ | 0.02 | 0.025 | 0.03 | 0.035 | 0.04 | 0.05 | 0.06 |
|---|---|---|---|---|---|---|---|
| construct-time (s) | 11.6533 | 12.4221 | 11.8358 | 11.6097 | 12.2473 | 12.2124 | 12.5851 |
| query-time (s) | 0.3504 | 0.4468 | 0.4446 | 0.4638 | 0.4650 | 0.4751 | 0.4874 |

