# OpenReview forum: "A Sublinear-Time Spectral Clustering Oracle with Improved Preprocessing Time"
_NeurIPS.cc/2023/Conference — NeurIPS 2023 poster_

### Official Review · Reviewer_Wy1e · 2023-07-05

**Soundness:** 3 good
**Presentation:** 3 good
**Contribution:** 3 good
**Rating:** 6
**Confidence:** 3

**Summary:**

This paper studies the problem of constructing a spectral clustering oracle with sublinear pre-processing and query time complexity. The paper introduces a new algorithm which improves on previous methods with respect to the running time at the expense of a slightly worse approximation guarantee. In contrast with previous methods, the new algorithm is more practical as demonstrated by experiments on synthetic data.

**Strengths:**

On the theoretical side, this paper introduces a new sublinear-time clustering oracle with guarantees which improve on the previous state of the art in terms of running time. The new proof techniques are introduced in quite a natural way and the intuition is clearly explained. An important contribution of this paper is that the proposed algorithm is practical and admits an implementation.

**Weaknesses:**

The theoretical improvements over the previous algorithms are quite small (in particular, the improvement over [30]), however I don't view this as a major weakness given that the new algorithm is more practical.

The statement of Theorem 1 is quite difficult to follow. For example, the parameter $\xi$ doesn't seem to be introduced or explained intuitively.

The experimental evaluation is quite limited - it does not include comparison with any other method, and does not report the running time of the algorithm. If the algorithm cannot be compared with that of [30] or [14], consider stating why this is the case in section 4.

**Questions:**

What is the purpose of $\xi$ in Theorem 1? As far as I can see from the Theorem statement, it only appears in the running times, and always in the denominator - why not always set it to $1$?

Is it possible to experimentally compare the algorithm with [30] and [14]?

### Typos
* Line 133 - space missing after '.'
* Line 144 - space missing after reference [7]
* Line 145 - Noted -> Note
* Line 246 - dose -> does
* Line 262 - space missing after 'Theorem'

**Limitations:**

Limitations are adequately addressed.

---

> ### Author Rebuttal · Authors · 2023-08-09
>
> Thank you very much for the careful reading and the helpful comments. We fixed the typos in the updated manuscript. We slightly summarized your questions and provided detailed answers below.
>
> **Question 1: The statement of Theorem 1 is quite difficult to follow. What is the purpose of $\xi$ in Theorem 1? It only appears in the running times, and always in the denominator - why not always set it to $1$?**
>
> (1) The primary intention behind the incorporation of $\xi$ into Theorem 1 was to more effectively illustrate the tradeoff between preprocessing/query time and the precision of the approximation for the dot product of spectral embeddings. This approximation forms a pivotal component of our clustering oracle. However, upon further consideration, we acknowledge that including $\xi$ in Theorem 1 is not essential (see below).
>
> (2) There is a typo for the upper bound of $\xi$ in the statement of Theorem 1, that is, the correct range of $\xi$ is $(\frac{1}{n^5},\frac{\gamma k}{5}]$ (instead of $\xi\in (\frac{1}{n^5},1)$). This range can be seen from the proof of Theorem 1 on page 8.
>
> (3) We concur that incorporating $\xi$ into Theorem 1 is not essential. Instead, it is sufficient to substitute $\xi$ with the upper bound $\frac{\gamma k}{5}$ and subsequently adjust the associated running times accordingly. In the next version of our work, we intend to implement this modification by directly replacing $\xi$ in Theorem 1 with the aforementioned upper bound.
>
> **Question 2: Is it possible to experimentally compare with the algorithm in [30] and [14]? If it cannot be compared with that of [30] or [14], consider stating why this is the case in section 4.**
>
> The algorithm in [14] is hard to implement.  As highlighted in our paper, the algorithm in [14] initially approximates the $k$ cluster centers by sampling around $O(1/\varepsilon\cdot k^4\log k)$ vertices, and subsequently undertakes the enumeration of approximately $2^{O(1/\varepsilon\cdot k^4\log^2 k)}$ potential $k$-partitions (Algorithm 10 in [14]). This enumeration process is extremely time-intensive and becomes impractical even for modest values of $k$. As suggested, we will explicitly state this factor in Section 4 to provide a clear rationale for our approach.
>
> Regarding the algorithm introduced in [30], it appears that its implementation is likely viable. However, our primary advancement over [30] is evident in the significantly reduced conductance gap we achieve. To thoroughly explore the comparative performance of our algorithm against [30], we aim to conduct new experiments. These experiments will focus on determining the specific graph ranges within which our algorithm outperforms [30]. In the upcoming version of our work, we are committed to implementing the algorithm from [30] and conducting the suggested experiments to provide a more comprehensive understanding of the algorithms' respective capabilities.
>
> **Question 3: The experimental evaluation is quite limited - it does not include comparison with any other method, and does not report the running time of the algorithm.**
>
> We are pleased to share that we have generated new experimental results that enrich our evaluation efforts. We kindly request your review of the attached PDF file within our rebuttal, where these results are detailed. Notably, these additions encompass experiments focusing on the algorithm's running time and its robustness, providing a more comprehensive assessment of its performance. Your consideration of these findings is greatly appreciated.
>
> * [Running time experiments] Please see Table 1 in the PDF file. The experimental results show that the running time of a single query is between 0.5-0.7 seconds, and the pre-processing time of our oracle is between 15.9-18.2 seconds.
>
> * [Robustness experiments] Please see Table 2 in the PDF file. We evaluate our robust algorithm on an SBM graph after deleting delNum edges in each cluster (chosen randomly), where delNum is a parameter. We found that as long as delNum is not too large, our oracle has a small misclassification error, e.g. for 50 edge deletions in each cluster, the error is only $0.8\textperthousand$.

---

> > ### Comment · Reviewer_Wy1e · 2023-08-16
> >
> > Many thanks for your detailed response. I am pleased to see that you will implement some of my suggested improvements. I will keep my current (positive) score.

---

### Official Review · Reviewer_2P47 · 2023-07-06

**Soundness:** 3 good
**Presentation:** 3 good
**Contribution:** 2 fair
**Rating:** 7
**Confidence:** 3

**Summary:**

This paper studies oracles for spectral graph clustering, i.e., local algorithms that answer membership queries for single nodes in a spectral clustering of a graph. There is a line of research on testing cluster structure in degree-bounded graphs, and recently, the learning version of the problem studied in this paper has become popular. Besides a result for robust clustering oracles by Peng, this works is closely related to a paper by Gluch et al. Compared to the work of Gluch et al., this submission improves the preprocessing time to $O(n^{1/2 + O(\epsilon / \phi^2)})$, which is better by a factor of approximately $2^{poly(k/e)}$ at the expsense of requiring a conductance gap of approximately $\Omega(1/poly(k))$, which is worse by approximately a $poly(k)/\log(k)$ factor, and a misclassification error of $O(poly(k) \epsilon)$, which is worse by approximately a $k / \log(k)$ factor. The misclassification error is the fraction of vertices that are assigned to the wrong cluster (compared to the ground truth clustering). The query time of the two algorithms is roughly the same. In a nutshell, the result in this submission trades an additional polynomial dependency in conductance gap and misclassification error against the removal of an exponential dependency in preprocessing time. The authors experimentally confirm the misclassification error and query complexity proven in their theorems.

This work builds up on the dot product oracle introduced by Gluch et al. The algorithm in the latter work estimates the means of the clusters (in an embedding space) and uses the dot product oracle to estimate the closest cluster center (mean) for a query node. The exponential preprocessing time arises from the former part. In the present submission, the authors propose an algorithm that doesn't estimate the cluster means, but compares the dot product between node embedding directly. Intuitively, if two nodes belong to the same cluster, they have a large dot product with their cluster mean, and so they should also have a large dot product with each other. This modification also results in the aforementioned trade off, i.e., the increased misclassification error and the stronger requirement on cluster separation.

The question answered by this paper arises naturally from the work of Gluch et al.: Do we need to compute the cluster means explicitly? While the answer may not be surprising (no, but there is a trade off), it requires some work to actually prove this, as the formal argument is not simple and obvious. The intention of the experiments is not clear to me, as there is no comparison with other algorithms, or insights how the theoretical algorithm needs to be modified and tuned for applications.

Rebuttal: Rating changed from weak accept to accept due to authors' rebuttal responses.

**Strengths:**

* The question explored in this paper is natural.
* The paper confirms an intuitive concept of spectral embeddings for clustering.

**Weaknesses:**

* The result in this paper is not very surprising or better than one would expect. It seems more like a reasonable trade off.
* The experiments seem currently not very useful.

**Questions:**

Note: It seems to me that a comparison of the conductance gap of your result and [14] is missing.

**Limitations:**

-

---

> ### Author Rebuttal · Authors · 2023-08-09
>
> Thank you very much for taking the time to review our paper.  We address your concerns in the following.
>
> **Summary: The intention of the experiments is not clear to me, as there is no comparison with other algorithms or insights on how the theoretical algorithm needs to be modified and tuned for applications.**
>
> **S1: The intention of our experiments.**
>
> (1) We aim to elevate our oracle beyond its theoretical framework. The outcomes of our experimentation, presented in Table 1 within our manuscript, distinctly demonstrate that our oracle's misclassification error remains notably minimal in instances where the input graph showcases an underlying latent cluster structure. This empirical validation reinforces the practical utility and efficacy of our oracle beyond theoretical conjecture.
>
> (2) We note that there is a tradeoff between computational cost and clustering quality. The main reason that we are adding the experiment on the query complexity, presented in Table 2 within our manuscript, is to show that for a small target misclassification error, our algorithms only require a **sublinear amount** of data, which is often critical when analyzing large social networks, since one typically does not have access to the entire network.
>
> **S2: Why there is no comparison with other algorithms?**
>
> We briefly comment on the implementation of the two most relevant sublinear-time clustering oracles given in [14] and [30].
>
> (1) Implementing the algorithm from [14] poses challenges. As highlighted in our paper, the algorithm in [14] initially approximates the $k$ cluster centers by sampling around $O(1/\varepsilon\cdot k^4\log k)$ vertices, and subsequently undertakes the enumeration of approximately $2^{O(1/\varepsilon\cdot k^4\log^2 k)}$ potential $k$-partitions (Algorithm 10 in [14]). This enumeration process is extremely time-intensive and becomes impractical even for modest values of $k$. We will explicitly state this factor in Section 4 to provide a clear rationale for our approach.
>
> (2) Regarding the algorithm introduced in [30], it appears that its implementation is likely viable. However, our primary advancement over [30] is evident in the significantly reduced conductance gap we achieve. To thoroughly explore the comparative performance of our algorithm against [30], we aim to conduct new experiments. These experiments will focus on determining the specific graph ranges within which our algorithm outperforms [30]. In the upcoming version of our work, we are committed to implementing the algorithm from [30] and conducting the suggested experiments to provide a more comprehensive understanding of the algorithms' respective capabilities.
>
> **S3: How the theoretical algorithm needs to be modified and tuned for applications?**
>
> In the upcoming version, we'll provide the following detailed steps to translate the theoretical algorithm into a practical one.
>
> To adapt the dot product oracle parameters, such as $t$ (random walk length), $s$ (sampling set size), and $R$ (number of random walks), we exploit the theoretical gap between intra-cluster and inter-cluster dot products in clusterable graphs. By constructing the oracle with various parameter settings and calculating intra-cluster and inter-cluster dot products, we generate density graphs. The setting with the most prominent gap in the density graph is selected (see Figure 1 in the attached PDF file within our rebuttal). Our misclassification error experiments adopt $t=25$, $s=20$, and $R=200$.
>
> Determining the appropriate threshold $\theta$ (lines 2, 8, 9 of Algorithm 1, and line 3 of Algorithm 2) is the next step. By observing the density graph linked to the chosen dot product oracle parameters, we identify the fitting $\theta$. In our misclassification error experiments, $\theta$ is set at $0.0006$ and $0.00053$ for $p=0.02$ and $0.0225$, respectively. For all other $p$ values in the range $[0.025, 0.06]$, $\theta$ is fixed at $0.0005$.
>
> Furthermore, for a WhichCluster($G,x$) query, the theoretical algorithm randomly selects an index if vertex $x$ belongs to multiple components of similarity graph $H$ (Algorithm 2). In practice, we return the index of the first component to which $x$ belongs. This adjustment enhances the algorithm's practical usability.
>
> **Weakness: The experiments seem currently not very useful.**
>
> Thanks for pointing out this. We are pleased to share that we have generated new experimental results that enrich our evaluation efforts. We kindly request your review of the attached PDF file within our rebuttal, where these results are detailed. Notably, these additions encompass experiments focusing on the algorithm's running time and its robustness, providing a more comprehensive assessment of its performance. Your consideration of these findings is greatly appreciated.
>
> * [Running time experiments] Please see Table 1 in the PDF file. The experimental results show that the running time of a single query is between 0.5-0.7 seconds, and the pre-processing time of our oracle is between 15.9-18.2 seconds.
>
> * [Robustness experiments] Please see Table 2 in the PDF file. We evaluate our robust algorithm on an SBM graph after deleting delNum edges in each cluster (chosen randomly), where delNum is a parameter. We found that as long as delNum is not too large, our oracle has a small misclassification error, e.g. for 50 edge deletions in each cluster, the error is only $0.8\textperthousand$.
>
> **Question: It seems to me that a comparison of the conductance gap of your result and [14] is missing.**
>
> Thanks for pointing out the missing comparison. We will fix this in our manuscript. We also provide the comparison of the conductance gap between the two results in the following.
>
> In [14], the conductance gap is $\varepsilon\ll O(\varphi^3/{\rm log}(k))$, and our conductance gap is $\varepsilon \ll O(\varphi^2/{\rm poly}(k))$.

---

> > ### Comment · Reviewer_2P47 · 2023-08-14
> >
> > Thank you for the detailed response!
> >
> > With the improvements outlined by the authors in all rebuttal comments and their commitment to incorporate them in the final version, I'm convinced that the paper becomes stronger, especially in the experimental part, and gives more significant insight into how the algorithm can be applied in practice. I bump my rating from weak accept to accept.

---

### Official Review · Reviewer_xqth · 2023-07-07

**Soundness:** 3 good
**Presentation:** 3 good
**Contribution:** 2 fair
**Rating:** 6
**Confidence:** 4

**Summary:**

This paper proposes a spectral clustering oracle with sublinear pre-processing time and query time. The query is in the form of $(G, x)$ where $G$ is a graph with underlying clusters and $x \in V$ is a vertex. The goal is to (1) construct the oracle efficiently, (2) report which cluster vertex $x$ belongs to efficiently.

Comparing to the previous work, the main contribution is improvement on the pre-processing time, which reduces exponential to polynomial on $O(k/\varepsilon)$, but blows up the misclassification error from $\log k$ to $\text{poly}(k)$, also slightly relaxing an assumption on the gap between inner and outer conductance.  The query time is asym-same.

The main technique is to replace the exhaustive search for each sampled vertex to decide a vertex $x$ belongs to a cluster with center $\mu$, with estimating the inner product of their spectral embeddings. It is proved the magnitude of the inner product roughly shows if two vertices belong to the same cluster.

**Strengths:**

- The paper is clearly written and well organized.
- The result is neat, the proposed algorithms are more easy to implement comparing to previous ones.
- I think for a theory paper, having experiments is always a plus. However, there is a mentality that either do it well or just don't do it. The experiments can be improved or at least clarified better.

**Weaknesses:**

- The major concern is that the contribution of the main result is quite limited. Yes, $O(k/\varepsilon)$ is an important factor, but it still in the $\tilde{\Omega}(\sqrt{n})$ regime, not to mention compromise on others. I actually like the robustness result better.
- On the experiments, if the authors want to keep and improve the section, I would suggest:
  - (1) Clarify the evaluation. I believe the theoretical result on the error is the number of query instances? Then the current report does not look like so, is it the fraction?
  - (2) The issue of query complexity is not mentioned before. It is out of blue. Explain why you do it.
  - (3) Add the robustness experiments.

**Questions:**

For the oracle results, there is another measure-of interest on the space. It would be good to report and compare with previous ones on this.

Just some curiosity on the lower bound. Is there any result stating something like: to obtain a small error, at least some space / time is needed?

Another suggestion is to give some conceptual description of good and bad vertices before the definition, maybe in the main technique part.

**Limitations:**

See above.

My reason to give the current assessment is mainly on the technical novelty limit.

---

> ### Author Rebuttal · Authors · 2023-08-09
>
> Thank you very much for taking your time to review our paper. We are happy to know that you like our robustness result. We address your concerns in the following.
>
> **The major concern is that the contribution of the main result is quite limited. Yes, polynomial on k/eps is an important factor, but it still in the Omega(\sqrt{n}) regime, not to mention compromise on others.**
>
> Since the running time of the spectral clustering oracle is $f(k) \Omega(\sqrt{n})$ regime, it seems that improving the f(k) factor from exp(k/eps) to poly(k/eps) is a modest improvement. We wish to highlight that achieving this improvement was not straightforward, besides that our algorithm is more practical and conceptually simpler. Our improvement necessitated the intricate application of spectral embeddings and the acquisition of certain insights, which appear deceptively simple *in hindsight*. Remarkably, during a presentation in the Bangalore Theory Seminars, one of the authors, M. Kapralov of [14],  raised an open question (refer to the segment commencing at 54:44 of the YouTube video): "if one can get polynomial dependency on $k$ or avoid enumerating over the candidate cluster centers..." and "get sublinear time for all $k=o(n)$".
>
> Our result made progress towards solving this question, and our algorithm does not enumerate over the candidate cluster centers and has sublinear running time for a much broader range of $k$ (with a slightly worse conductance gap than [14]).
>
> **Clarify the evaluation. I believe the theoretical result on the error is the number of query instances? Then the current report does not look like so, is it the fraction?**
>
> Thanks, we will clarify this in the upcoming version. In the context of our study, the misclassified error pertains to the **fraction** of inaccurately categorized vertices within each cluster. In Theorem 1, we employ the assertion $|U_{\pi(i)}\triangle C_i|\le O({\rm poly}(k\cdot\varepsilon))|C_i|$ as a metric for quantifying this error. This can also be equivalently expressed as the fraction of misclassified vertices being $O({\rm poly}(k\cdot \varepsilon))$ within each individual cluster $C_i$. In our experiments, we directly utilize this error fraction within our report, as exemplified in Table 1 in our submitted manuscript.
>
> **The issue of query complexity is not mentioned before. It is out of blue. Explain why you do it.**
>
> We note that there is a tradeoff between computational cost and clustering quality. The main reason that we are adding the experiment on the query complexity is to show that for a small target misclassification error, our algorithms only require a **sublinear amount** of data, which is often critical when analyzing large social networks, since one typically does not have access to the entire network.
>
> **Add the robustness experiments.**
>
> Please see the attached PDF file (Table 2) within our rebuttal for our additional experiments on the robustness of our spectral clustering oracle.
>
> For example, we evaluate our robust algorithm on an SBM graph after deleting delNum edges in each cluster (chosen randomly), where delNum is a parameter. We found that as long as delNum is not too large, our oracle has a small misclassification error, e.g. for 50 edge deletions in each cluster, the error is only $0.8\textperthousand$.
>
> **For the oracle results, there is another measure-of interest on the space. It would be good to report and compare with previous ones on this.**
>
> Thanks for the suggestion. Upon further consideration, we find that our data structure uses much smaller space than the one in [14] for some interesting regime of parameters. We will add the following clarification in the next version of our manuscript.
>
> The space complexity of our data structure is $O\left({\rm poly}(k)\cdot n^{1/2+O(\varepsilon/\varphi^2)}\cdot {\rm poly}(\log n)\right)$, while the oracle in [14] needs $O\left({\rm poly}(k/\varepsilon)\cdot n^{1/2+O(\varepsilon/\varphi^2)}\cdot {\rm poly}(\log n)\right)$. That is, the first term in our space complexity does **not** depend on $\varepsilon$ in comparison to the one in [14]. The primary factor behind this enhancement lies in our ability to approximate the dot product of spectral embeddings with only an additive error of $1/{\rm poly}(k)$, whereas [14] necessitates achieving an additive error of ${\rm poly}(\varepsilon/\varphi)$.
>
> It's worth highlighting that our space complexity significantly outperforms that of [14], particularly in cases where $k$ is fixed and $\varepsilon$ takes on exceptionally small values, such as $\varepsilon=1/n^{c}$, for sufficiently small constant $c>0$.
>
> **Is there any result stating something like: to obtain a small error, at least some space/time is needed?**
>
> Thanks for the question. We will add the following clarification in the next version of our manuscript.
>
> We note that there is no clustering oracle that allows both $o(\sqrt{n})$ preprocessing time and $o(\sqrt{n})$ query time. The reason is as follows: in [GR97], the authors show that one needs $\Omega(\sqrt{n})$ queries to distinguish an expander with $n$ vertices from a graph that is a union of two disjoint expanders each with $n/2$ vertices. Note that if there exists an oracle with both $o(\sqrt{n})$ preprocessing time and $o(\sqrt{n})$ query time, then one can use this oracle to distinguish the above two graphs, by checking if the corresponding similarity graphs have one or two connected components. This  will be a contradiction to the $\Omega(\sqrt{n})$ lower bound [GR97].
>
> Reference: [GR97] O. Goldreich, D. Ron. Property testing in bounded degree graphs. STOC 1997.
>
> **Another suggestion is to give some conceptual description of good and bad vertices before the definition, maybe in the main technique part.**
>
> Thanks, we will give a brief informal description of good and bad vertices in Section 3.1 (our techniques) as you suggested.

---

> > ### Comment · Reviewer_xqth · 2023-08-21
> >
> > Thank the authors for the detailed response. I am happy to see that the space complexity also roughly shaves off an $\frac{1}{\varepsilon}$ factor. Together with the lower bound, the results make this manuscript a more complete work. As for the experiments, please just make sure the setup is clearly explained.
> >
> > Since my major concern is on the technical depth and the authors have shown this improvement answers to a standing open problem. I am raising the score to 6. Thank you!

---

### Author Rebuttal · Authors · 2023-08-10

In response to the review comments, we have incorporated two additional experimental outcomes and included a figure outlining the parameter tuning process for the theoretical algorithm. We invite you to refer to the attached PDF file for both the experimental results and the figure explanation. Your review of the attached PDF file is greatly appreciated. Thank you.

---

> ### Comment · Reviewer_2P47 · 2023-08-11
>
> Thank you for the detailed rebuttal responses! I acknowledge that a rebuttal does not allow for fully fleshed out answers to each question. For some updates that you announced for the "upcoming version", I may think that it is their existence that contributes most to the scientific thoroughness of a paper, not their exact outcome (which would be unknown beforehand). Therefore, may I ask whether this upcoming version is a commitment to include these changes in the final version?

---

> > ### Author Response · Authors · 2023-08-12
> >
> > Thank you for your understanding and consideration of our rebuttal responses. Indeed, the primary intention behind mentioning updates for the "upcoming version" is to enhance the completeness and robustness of our research.
> >
> > In response to your question, we confirm that the outlined updates are indeed commitments that will be integrated into the final version of the paper. We are pleased to share that a significant portion of these updates, as detailed in our previous response, have already been successfully incorporated. This includes the changes presented in the attached PDF file within our rebuttal.

---

### Decision · Program_Chairs · 2023-09-21

**Decision:**

Accept (poster)

**Comment:**

The paper received universally positive reviews -- it answers an interesting theoretical question -- improving the runtime of prior work on sublinear time spectral clustering oracles from exponential in the number of clusters and error parameter to polynomial, at the cost of requiring an exponentially larger (log k -> poly(k)) gap between the clusters.

While the paper is largely theoretical, one major concern was the experimental section. The authors have given expanded results during the rebuttal, which we hope will be included in the final version of the paper. We encourage the authors to address additional gaps, notably the lack of comparison to any other approaches. Or, at least to clearly mention the limitations of the experiments in the paper.